# Single-cell transcriptomics of human cholesteatoma identifies an activin A-producing osteoclastogenic fibroblast subset inducing bone destruction

Kotaro Shimizu [1,2,3], Junichi Kikuta [1,2,4] ✉, Yumi Ohta[3], Yutaka Uchida [1,2], Yu Miyamoto[1,2], Akito Morimoto[1,2], Shinya Yari [1,2], Takashi Sato[3], Takefumi Kamakura[3], Kazuo Oshima[3], Ryusuke Imai[3], Yu-Chen Liu [5,6], Daisuke Okuzaki [5,6], Tetsuya Hara[7], Daisuke Motooka[5,6], Noriaki Emoto [7], Hidenori Inohara[3] & Masaru Ishii [1,2,4] ✉

Cholesteatoma, which potentially results from tympanic membrane retraction, is characterized by intractable local bone erosion and subsequent hearing loss and brain abscess formation. However, the pathophysiological mechanisms underlying bone destruction remain elusive. Here, we performed a single-cell RNA sequencing analysis on human cholesteatoma samples and identify a pathogenic fibroblast subset characterized by abundant expression of inhibin βA. We demonstrate that activin A, a homodimer of inhibin βA, promotes osteoclast differentiation. Furthermore, the deletion of inhibin βA /activin A in these fibroblasts results in decreased osteoclast differentiation in a murine model of cholesteatoma. Moreover, follistatin, an antagonist of activin A, reduces osteoclastogenesis and resultant bone erosion in cholesteatoma. Collectively, these findings indicate that unique activin A-producing fibroblasts present in human cholesteatoma tissues are accountable for bone destruction via the induction of local osteoclastogenesis, suggesting a potential therapeutic target.

Cholesteatoma is a type of chronic middle ear inflammation that expands with bone erosion, destroying temporal bone structures and causing symptoms such as hearing loss, dizziness, facial paralysis, and meningitis[1]. Furthermore, cholesteatoma constitutes an epidermal cyst that arises from the epithelial layer of the tympanic membrane and is composed of three layers: matrix, composed of the keratinized stratified squamous epithelium; perimatrix, the surrounding layer of the matrix that contacts temporal bone and contains collagen fibers, fibroblasts, and endothelial cells; and cystic components that form the most internal layer, containing keratin debris and necrotic tissue shed from the matrix[2]. Multiple mechanisms have been proposed to explain bone erosion in cholesteatoma, including osteoclast activation[3,4], acid

[1]Department of Immunology and Cell Biology, Graduate School of Medicine and Frontier Biosciences, Osaka University, Suita, Osaka 565-0871, Japan. [2]WPI-Immunology Frontier Research Center, Osaka University, Suita, Osaka 565-0871, Japan. [3]Department of Otorhinolaryngology-Head and Neck Surgery, Graduate School of Medicine, Osaka University, Suita, Osaka 565-0871, Japan. [4]Laboratory of Bioimaging and Drug Discovery, National Institutes of Biomedical Innovation, Health and Nutrition, Ibaraki, Osaka 567-0085, Japan. [5]Genome Information Research Center, Research Institute for Microbial Diseases, Osaka University, Suita, Osaka 565-0871, Japan. [6]Laboratory of Human Immunology (Single Cell Genomics), WPI-Immunology Frontier Research Center, Osaka University, Suita, Osaka 565-0871, Japan. [7]Laboratory of Clinical Pharmaceutical Science, Kobe Pharmaceutical University, Higashinada, Kobe 658-8558, Japan. ✉e-mail: jkikuta@icb.med.osaka-u.ac.jp; mishii@icb.med.osaka-u.ac.jp

lysis[5], pressure necrosis[3,6], inflammatory mediators[7–9], enzymatic mediators[10,11], and combinations of ≥2 of these mechanisms. Nevertheless, the mechanism that underlies local bone destruction in cholesteatoma has not been elucidated. The only effective treatment at present is complete surgical excision, but the rate of postoperative recurrence remains unsatisfactory[12].

A previous study showed that numerous osteoclasts were observed on the eroded bone surfaces adjacent to cholesteatomas, compared to unaffected areas, and that fibroblasts in the cholesteatoma perimatrix express receptor activator of NF-κB ligand (RANKL), a protein essential for osteoclast differentiation and function[4,13]. Because multiple subtypes of fibroblasts have been reported in other inflammatory diseases, such as rheumatoid arthritis[14], it is possible that cholesteatoma also contains several subtypes of fibroblasts. However, previous studies have not provided an overview of cell types and subsets present in cholesteatoma. Here, we performed single-cell RNA sequencing (scRNA-seq) analysis of human cholesteatoma specimens to clarify the contributions of these cells to bone destruction in cholesteatoma. The results showed that cholesteatoma perimatrix fibroblasts express high levels of activin A, a secreted protein that acts in cooperation with RANKL to induce mature osteoclast formation. We conducted a detailed assessment of the relationship between activin A

and bone erosion in cholesteatoma, and the results showed that activin A is a potential therapeutic target for cholesteatoma.

## Results

### Single-cell RNA sequencing analysis identified a cholesteatoma-specific pathogenic fibroblast subset

To elucidate the mechanism underlying bone erosion in cholesteatoma, we performed scRNA-seq analysis using human cholesteatoma tissues surgically resected from patients. As there are no "normal" tissues in the middle ear that correspond to cholesteatoma, retroauricular skin at the incision site was used as a control for cholesteatoma. Previously, we reported the involvement of fibroblasts in bone erosion in cholesteatoma mediated via RANKL signaling[4]; therefore, we focused on pathogenic nonimmune cells, such as fibroblasts or keratinocytes, in the present study. Both cholesteatoma and control skin specimens were enzymatically digested and sorted into CD45⁻ live cell populations (Fig. 1a). Then, we applied scRNA-seq to the collected cells and analyzed the data sets. We first carried out quality control to exclude poor-quality data; we obtained a reliable data set consisting of 8357 cells from the cholesteatoma samples and 10,916 cells from the control skin samples obtained from three patients. Next, we performed clustering analysis and visualized the results in uniform

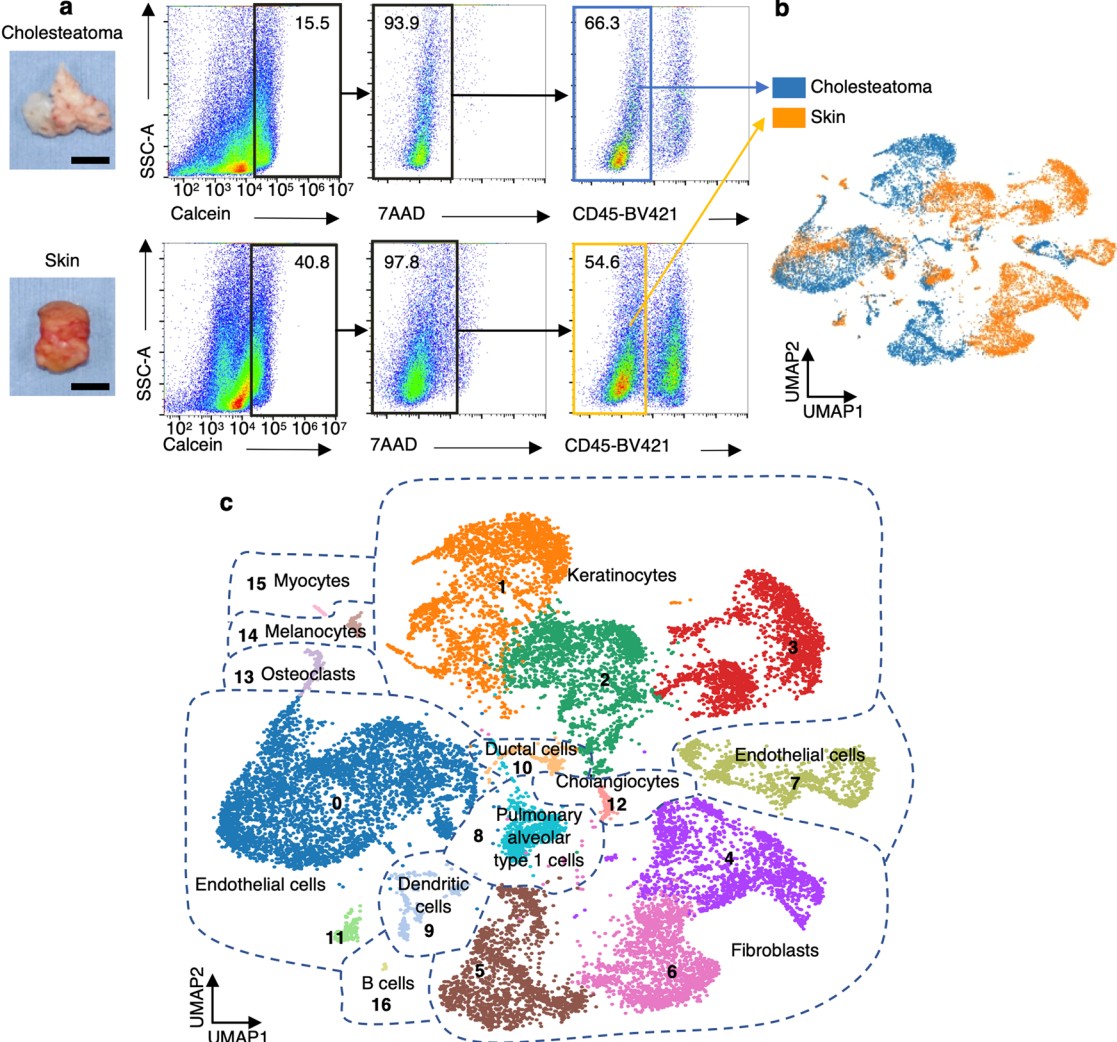

**Fig. 1 | scRNA-seq analysis of human cholesteatoma and skin specimens.**
**a** Representative gating strategies used in cholesteatoma and skin samples. Live (calcein⁺ AAD⁻) CD45⁻ cells. Scale bars: 5 mm. **b** UMAP plot of scRNA-seq data from 19,273 cells labeled by sample condition. Samples were obtained from three pairs of cholesteatoma and control skin samples labeled according to sample condition. **c** UMAP plot of scRNA-seq data labeled according to cell type identified in PanglaoDB. The main cell types were keratinocytes, fibroblasts, and endothelial cells.

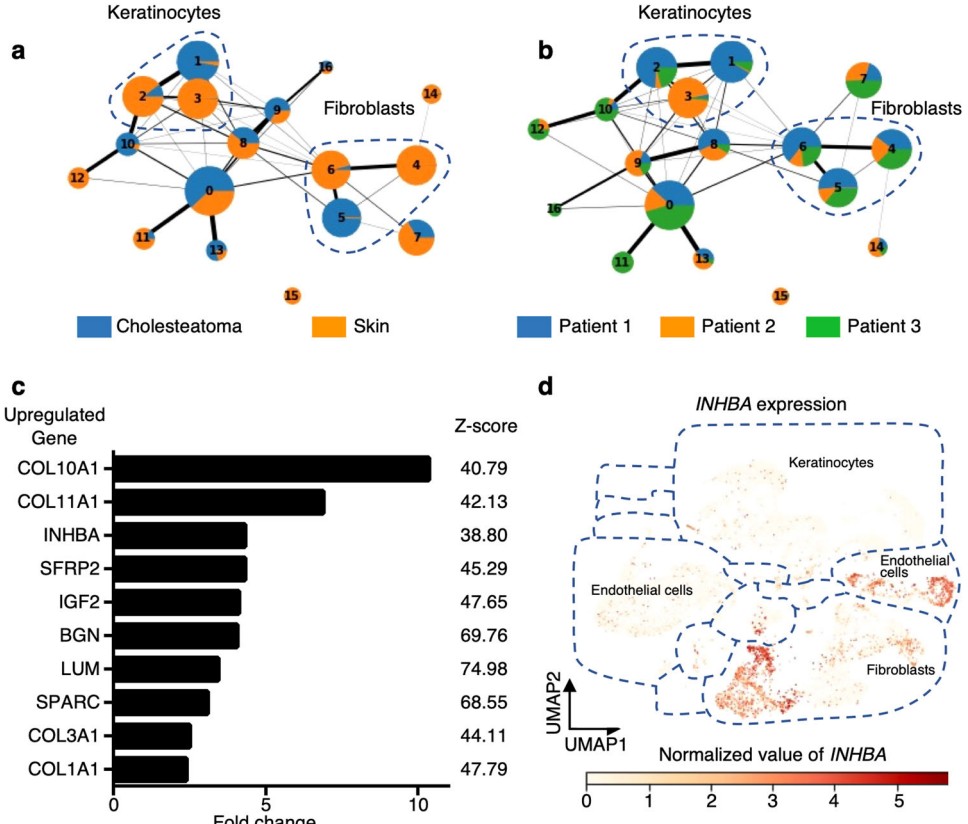

**Fig. 2 | Identification of cholesteatoma fibroblasts and *INHBA* upregulation in cholesteatoma fibroblasts. a** The proportions of sample conditions in each cluster identified by scRNA-seq. The proportion of cholesteatoma varied among clusters. Clusters consisting of cholesteatoma alone were considered cholesteatoma-specific clusters. **b** Proportion of patients in each cluster identified by scRNA-seq.

The proportion of patients varied among clusters. Clusters consisting of one patient were considered clusters with large sample biases. **c** Genes upregulated in cholesteatoma fibroblasts compared to control fibroblasts. Genes with top 10 z-scores are shown in order of fold change. **d** *INHBA* upregulation in cholesteatoma fibroblast clusters.

manifold approximation and projection (UMAP). As shown in Fig. 1b, both cholesteatoma and control dermal cells showed several clusters (blue: cholesteatoma; orange: control skin; Supplementary Fig. S1 shows color coding by the patient). Our clustering method classified the samples into 17 clusters, and we annotated the cell types based on marker gene expression (Fig. 1c). We proposed that the data sets consisted of 11 distinct cell types, including keratinocytes, fibroblasts, endothelial cells, and osteoclasts (Fig. 1c; Supplementary Figs. S2 and S3). Because keratinocytes, fibroblasts, and endothelial cells were major populations in the data sets, subsequent analysis focused on these cells.

To identify cholesteatoma-specific keratinocytes, fibroblasts, and endothelial cell clusters, we evaluated the proportions of cholesteatoma-derived cells and normal dermal cells in each cluster. The keratinocytes in cluster 1 and fibroblasts in cluster 5 were strongly biased toward the cholesteatoma samples, whereas other clusters, including endothelial cell clusters 0 and 7, were unbiased (Fig. 2a). We also examined bias between patients in keratinocyte cluster 1 and fibroblast cluster 5. The cell proportion in fibroblast cluster 5 was unbiased toward any particular patient, whereas keratinocyte cluster 1 was strongly biased toward patient 1 (Fig. 2b). We also confirmed that cluster 5 in fibroblasts had higher condition specificity and lower sample bias (Supplementary Fig. S4a–c). Taken together, our data suggest that cluster 5 represented a cholesteatoma-specific pathogenic fibroblast subset. To investigate the characteristics of the cholesteatoma-specific pathogenic fibroblast subset, we next compared gene expression in the subsets with clusters 4 and 6 that were biased toward normal dermal fibroblasts. The upregulated genes in the cholesteatoma-specific subset with the top 10 z-scores are shown in

Fig. 2c. Most of the upregulated genes were related to connective tissues of the extracellular matrix, including *COL10A1*, *Col11A1*, and *COL3A1*. We also found several encoding-secreted proteins, including *INHBA* (inhibin βA), *SFRP2*, and *IGF2*. Among the genes encoding secreted proteins, we focused on *INHBA*, which had the largest fold change in expression (4.32-fold). We visualized *INHBA* expression levels across all clusters in UMAP and confirmed that *INHBA* was highly expressed in cluster 5, the cholesteatoma-specific fibroblast subset, although *INHBA* expression was also seen in endothelial cells, albeit at low levels (Fig. 2d). These data prompted us to investigate further whether *INHBA* expression in cholesteatoma-specific fibroblasts was involved in osteoclastogenesis in cholesteatoma.

## Fibroblast subclustering analysis and trajectory mapping identified pathogenic fibroblasts with high *INHBA* expression in cholesteatoma

To infer the process of development from normal to disease-specific fibroblasts, we classified each fibroblast cluster into smaller sub-clusters and performed pseudotime trajectory analysis using Monocle 3 (Fig. 3a, b). Subcluster analysis revealed that the fibroblasts contained 15 subclusters. Cholesteatoma-specific fibroblasts were divided into five subclusters designated as 1, 7, 8, 10, and 11 (Fig. 3a). We identified the top 10 marker genes indicating the differentiation status of fibroblasts, then selected the *PI16* gene (described as a universal fibroblast marker in a previous study[15]) (Supplementary Fig. S5a). Subcluster 9 was speculative as the origin of fibroblasts for pseudotime trajectory analysis because cells in this subcluster showed high levels of *PI16* expression (Supplementary Fig. S5b). Pseudotime trajectory analysis suggested that the cholesteatoma-specific fibroblasts

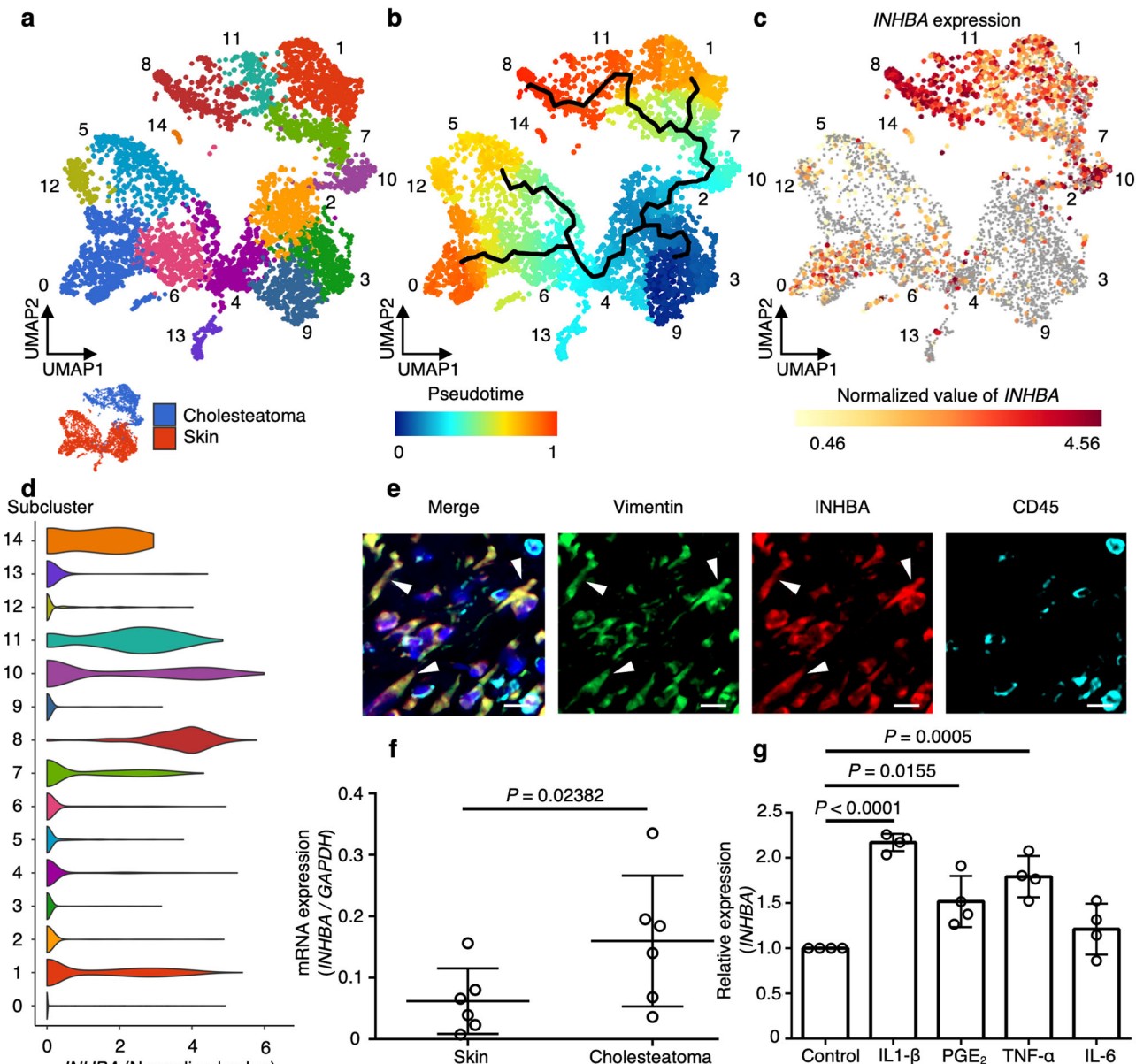

**Fig. 3 | Subclustering and pseudotime analysis of cholesteatoma fibroblasts and confirmation of *INHBA* expression in human cholesteatoma.**
**a** Subclustering analysis of fibroblasts. Populations shown at the upper right are cholesteatoma fibroblasts. Populations shown at the lower left are skin fibroblasts. Cholesteatoma fibroblasts are labeled blue, and skin fibroblasts are labeled red at the lower left. Cholesteatoma fibroblasts were associated with five subclusters labeled 1, 7, 8, 10, and 11. **b** Trajectory mapping performed using Monocle 3. Undifferentiated and differentiated cells are labeled blue and red, respectively. The most differentiated cells were observed in cholesteatoma (labeled red). The differentiated cells were identical to cholesteatoma fibroblasts in subcluster 8.
**c** *INHBA* expression in fibroblasts in UMAP. Cholesteatoma fibroblasts showed high levels of *INHBA* expression. The area of high *INHBA* expression was identical to the area in cholesteatoma fibroblasts in subcluster 8. **d** Violin plots of *INHBA* expression in each cluster. Subcluster 8 shows a high *INHBA* normalized value.

**e** Immunofluorescence staining of cholesteatoma perimatrix. Colocalization of vimentin and INHBA immunofluorescence showed that fibroblasts in the perimatrix expressed INHBA. CD45 was not colocalized with activin A. Leukocytes exhibited no activin A expression. INHBA was labeled with Alexa Fluor 568, vimentin was labeled with Alexa Fluor 488, and CD45 was labeled with Alexa Fluor 647. Scale bars: 10 μm.
**f** *INHBA* mRNA expression level, corrected for *GAPDH* mRNA expression, was significantly higher in cholesteatoma fibroblasts than in control skin fibroblasts ($n = 6$). Data represent means ± SDs. **g** IL-1β, PGE$_2$, and TNF-α promoted *INHBA* expression in human primary skin fibroblasts. IL-6 did not promote *INHBA* expression in ear pinna-derived fibroblasts ($n = 4$). The exact *p*-value between the control and IL-1β is 0.0000003. Data represent means ± SDs. Statistical significance was determined using the ratio paired two-tailed *t*-test (**f**) and two-tailed unpaired *t*-test (**g**). Source data are provided as a Source Data file.

differentiated from subcluster 9 toward subcluster 8 via subclusters 3, 2, 10, 7, and 11 (Fig. 3b). The marker gene *PI16* was downregulated after differentiation (Supplementary Fig. S5b). These results suggest subcluster 9 as the potential origin of fibroblasts; however, a comprehensive tissue localization analysis and lineage tracing are necessary to definitively establish the source of fibroblasts. Subcluster 8 was composed of the most differentiated cholesteatoma-specific fibroblasts

and exhibited markedly higher *INHBA* expression levels than the other subclusters in the cholesteatoma-specific fibroblasts (Fig. 3b–d, Supplementary Fig. S5c). This was consistent with the results of Gene Ontology (GO) biological pathway analysis, which showed that subcluster 8 was correlated with multicellular organism development relative to other cholesteatoma-specific fibroblast subpopulations (Supplementary Fig. S5d). We also identified cell-surface markers of

cholesteatoma-specific fibroblast subcluster 8. We investigated the top 100 genes upregulated in subcluster 8, then selected cell surface or transmembrane proteins. Among these genes, we identified *TMEM119* and *PMEPA1* as candidate surface markers (Supplementary Fig. S6). Taken together, these observations indicated that cholesteatoma fibroblasts expressed *INHBA* at significantly higher levels than control dermal fibroblasts; they suggested that *INHBA* has potential as a marker gene to identify the more differentiated cholesteatoma-specific fibroblast subset.

### *INHBA* gene and activin A expression in human cholesteatoma specimens

Next, we examined INHBA expression in cholesteatoma fibroblasts by immunostaining of cholesteatoma sections with anti-INHBA antibodies. Cholesteatoma sections exhibited colocalization of INHBA and vimentin, a marker for fibroblasts, indicating that INHBA was present at the protein level in the cholesteatoma fibroblasts (Fig. 3e). We also confirmed *INHBA* expression at the mRNA level in the cholesteatoma fibroblasts using droplet digital polymerase chain reaction (ddPCR). We collected fibroblasts with the following markers, live (calcein⁺) and lineage-negative (CD31⁻ CD45⁻ E-cadherin⁻) cells, and performed ddPCR (Supplementary Fig. S7a). The expression of *INHBA* mRNA normalized relative to glyceraldehyde-3-phosphate dehydrogenase (*GAPDH*) mRNA was significantly higher in cholesteatoma fibroblasts than in control dermal fibroblasts (Fig. 3f). We also confirmed the expression of RANKL which is the most important molecule for osteoclastogenesis. Consistent with a previous study[4], immunofluorescence staining showed the coexpression of vimentin and RANKL, indicating that RANKL was expressed in fibroblasts in cholesteatoma (Supplementary Fig. S7b). The level of *RANKL* expression normalized relative to that of *GAPDH* was significantly higher in cholesteatoma than in the control dermis (Supplementary Fig. S7c).

Next, we attempted to identify signals that induced *INHBA* expression in the fibroblasts. As cholesteatoma specimens have been shown to express proinflammatory cytokines, such as interleukin-1 beta (IL-1β), interleukin-6 (IL-6), tumor necrosis factor-alpha (TNF-α), and prostaglandin E₂ (PGE₂)[4], we examined whether the proinflammatory cytokines promote *INHBA* expression in fibroblasts. Human primary skin fibroblasts were treated with proinflammatory cytokines, and the *INHBA* mRNA expression level was analyzed by reverse transcription PCR (RT-PCR). Among the proinflammatory cytokines, IL-1β, PGE₂, and TNF-α significantly promoted *INHBA* expression in human primary skin fibroblasts, suggesting the significance of proinflammatory cytokine signals for *INHBA* expression and pathological fibroblast differentiation (Fig. 3g). Additionally, universal fibroblast marker gene *PI16* mRNA expression was decreased by proinflammatory cytokines (Supplementary Fig. S7d), suggesting that inflammatory cytokines cause differentiation from human skin fibroblasts with general characteristics to cholesteatoma-specific fibroblasts expressing high levels of *INHBA*.

### Functional analysis of activin A in vitro

Next, we analyzed the ability of INHBA to promote osteoclast differentiation in vitro. *INHBA* encodes the A subunit of inhibin β, which forms dimers via a disulfide bridge with other inhibin subunits. INHBA forms activin A (a homodimer of INHBA), activin AB (a heterodimer of INHBA and INHBB), and inhibin A (a heterodimer of INHBA and INHA)[16]. Therefore, we compared the expression patterns of *INHBA*, *INHBB*, and *INHA* in human cholesteatoma fibroblasts by ddPCR. The results showed that *INHBA* was significantly upregulated relative to *INHBB* and *INHA* in cholesteatoma fibroblasts (Supplementary Fig. S8a). Because no upregulation of *INHBB* and *INHA* was observed in cholesteatoma fibroblasts, we concluded that upregulated INHBA in cholesteatoma-specific fibroblasts forms a homodimer, activin A. SFRP2 and IGF2 (secreted proteins that are upregulated in cholesteatoma fibroblasts except activin A) did not promote osteoclast differentiation; therefore, we used activin A protein for further in vitro experiments (Supplementary Fig. S8b–e). Because activin A enhances RANKL-induced osteoclast differentiation[16] through an interaction between c-Fos and Smad2/3[17], we first investigated the synergistic effects of activin A and RANKL on osteoclastogenesis in vitro. To evaluate the osteoclast differentiation, we used double-reporter CX₃CR1-enhanced green fluorescent protein (EGFP) knock-in (CX₃CR1-EGFP) mice[18] and tartrate-resistant acid phosphatase (TRAP) promoter-dependent tdTomato-expressing (TRAP-tdTomato) mice[19]. CX₃CR1, a fractalkine receptor, is considered a marker of monocyte-lineage cells, including osteoclast precursors; TRAP is a marker of mature osteoclasts. Bone marrow-derived macrophages derived from CX₃CR1-EGFP/TRAP-tdTomato mice were cultured in the presence or absence of activin A and/or RANKL; subsequently, the area of TRAP⁺ osteoclasts was quantified. We found that activin A did not induce TRAP⁺ mature osteoclasts, whereas the combination of activin A with RANKL synergistically increased the number of mature TRAP⁺ osteoclasts (Fig. 4a, b, Supplementary Fig. S9a–c). An inhibitor of the activin A receptor ALK-4 suppressed the synergistic effect of activin A, as well as the basal effect of RANKL on osteoclastogenesis (Supplementary Fig. S9d, e), suggesting that activin A is a key cofactor for RANKL-dependent osteoclastogenesis.

### Functional analysis of activin A in vivo

Next, we examined the in vivo role of activin A in osteoclastogenesis in cholesteatoma using an experimental mouse model of cholesteatoma that we developed previously[20]. Keratinocytes were isolated from the epidermis and primary fibroblasts were isolated from the dermis of control or INHBA-deficient mouse ear pinnae. To elucidate the expression of *INHBA*, *INHBB*, and *INHA* in cholesteatoma mouse model fibroblasts, we collected fibroblasts from the cholesteatoma mouse model and control mouse ear pinnae via cell sorting (Supplementary Fig. S10a). The expression of *INHBA* was significantly upregulated in fibroblasts from a mouse model of cholesteatoma compared to controls (Supplementary Fig. S10b). In addition, the levels of *INHBB* and *INHA* expression were lower than the level of *INHBA* in those fibroblasts (Supplementary Fig. S10c). These results suggest that INHBA forms dimers and activin A secretion is elevated in the cholesteatoma mouse model. The ear pinnae fibroblasts were cultured with 4-hydroxytamoxifen (4OHT) to block *INHBA* expression, and then the fibroblasts and keratinocytes were mixed at a 1:5 ratio and transplanted under the calvarial periosteum of TRAP-tdTomato mice (Fig. 4c). The number of osteoclasts formed on the parietal bone surface was significantly smaller in the *INHBA*-deficient fibroblast transplant group than in the control fibroblast transplant group (Fig. 4d, e). Furthermore, the application of the activin A antagonist follistatin (FST) suppressed osteoclastogenesis compared to the vehicle-injected group (Fig. 4f, g). Our data indicated that activin A plays a critical role in cholesteatoma-related osteoclastogenesis in vivo.

## Discussion

We performed a scRNA-seq analysis of human cholesteatoma samples and showed that proinflammatory cytokines induced a unique pathogenic fibroblast subset with abundant expression of INHBA/activin A. Furthermore, INHBA/activin A from the pathogenic fibroblast was a key molecule for ectopic osteoclastogenesis in a cholesteatoma mouse model. Based on our results, we proposed a molecular mechanism for bone destruction in cholesteatoma, as shown in Fig. 5. Proinflammatory cytokines, such as IL-1β and PGE₂, induce activin A-expressing pathogenic fibroblasts and activin A act in conjunction with RANKL to promote ectopic osteoclastogenesis. Although a number of hypotheses have been proposed for the mechanism of bone erosion in cholesteatoma, our results support an osteoclast activation model.

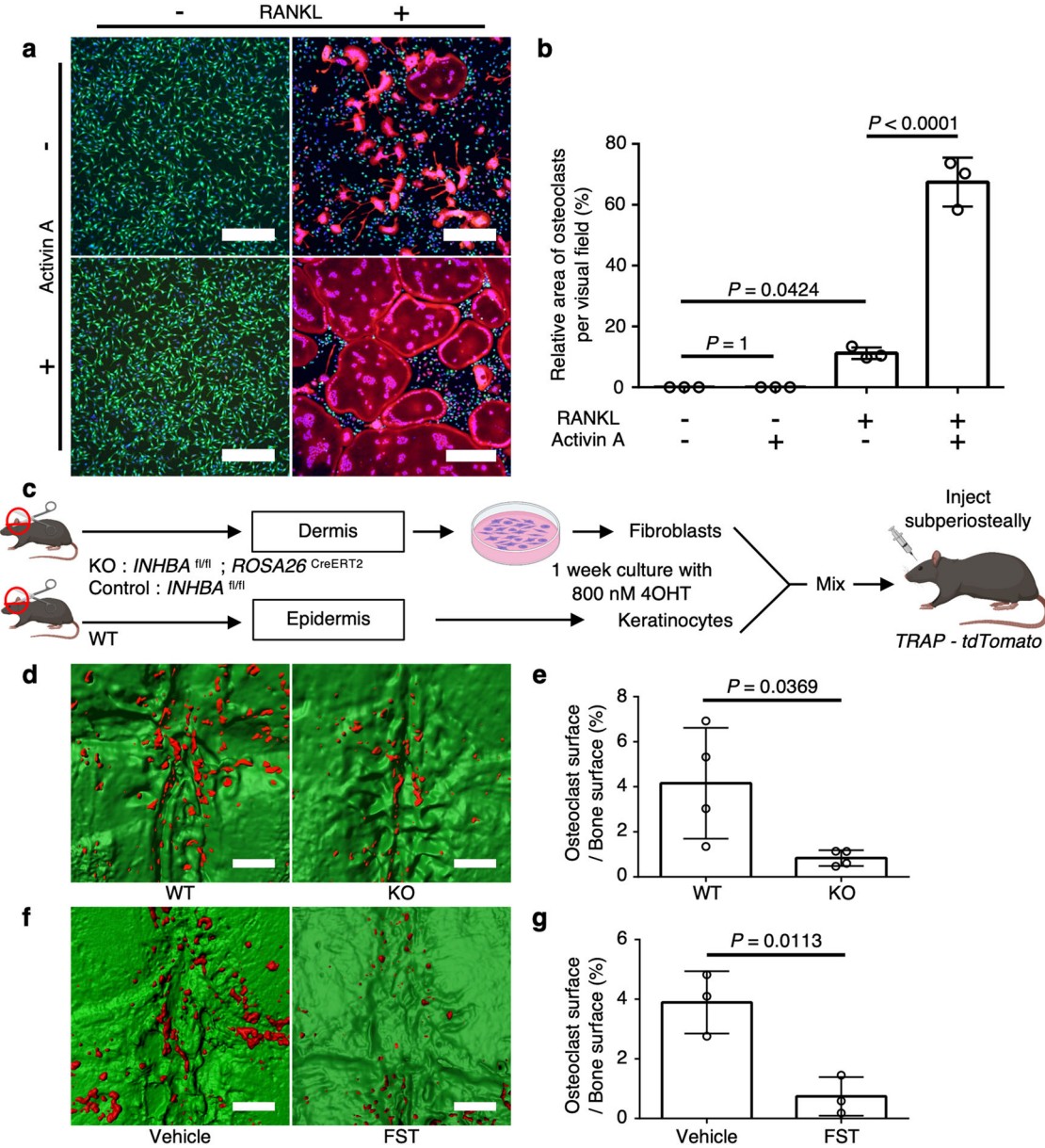

**Fig. 4 | Functional analysis of activin A in vitro and in vivo. a** Representative images of osteoclast differentiation in vitro. Bone marrow macrophages derived from CX₃CR1-EGFP/TRAP-tdTomato mice were cultured with 10 ng mL⁻¹ M-CSF. RANKL (50 ng mL⁻¹) and activin A (5 ng mL⁻¹) were added either alone or in combination. Green: CX₃CR1-EGFP⁺ cells; red: TRAP-tdTomato⁺ cells; blue: nuclei. Scale bars: 300 μm. **b** Quantification of TRAP-positive area within the visual field (*n* = 3 per group). Data represent means ± SDs. The exact *p*-value between RANKL (+) group and RANKL (+) activin A (+) group is 0.0000008. **c** Schematic representation of the method used to generate a mass lesion composed of ear pinna-derived keratinocytes and fibroblasts. **d** Representative surface images of TRAP-positive osteoclasts induced on the parietal bone surface. TRAP-positive cells were labeled with tdTomato, and parietal bones were labeled with Alexa Fluor 488. INHBA

inhibition in fibroblasts reduced osteoclast formation on the parietal bone surface. Red, TRAP-tdTomato⁺ cells; green, parietal bone surface. Scale bars: 500 μm. **e** Quantification of TRAP-positive area on the parietal bone surface under the cholesteatoma mass. WT cholesteatoma model (*n* = 4), INHBA inhibition in fibroblasts (*n* = 4). **f** Representative surface images of TRAP-positive osteoclasts induced on the parietal bone surface. Follistatin (FST) treatment reduced osteoclast formation on the parietal bone surface. Scale bars: 500 μm. **g** Quantification of TRAP-positive area on the parietal bone surface under the cholesteatoma mass. Vehicle treatment (*n* = 3), FST treatment (*n* = 3). Statistical significance was determined by one-way ANOVA with Tukey's post hoc multiple comparison test (**b**) and two-tailed unpaired *t*-test (**e, g**). Data represent the mean ± SD for each group. Symbols represent individual mice. Source data are provided as a Source Data file.

Our results suggest that the expression of inflammatory cytokines is an important step for ectopic osteoclastogenesis. Cholesteatoma is caused by chronic inflammation of the middle ear, and human cholesteatoma tissue secretes proinflammatory cytokines, such as IL-1β, IL-6, TNF-α, and PGE₂[4,21]. Among such proinflammatory cytokines, we showed that IL-1β, PGE₂, and TNF-α induced *INHBA* expression in fibroblasts from the primary skin of humans. Our results are consistent with previous reports that INHBA expression is induced by IL-1β through the p38 MAPK and MEK/ERK pathways and by PGE₂ through

the PKC and MEK/ERK pathways[22]. Furthermore, INHBA expression is induced by TNF-α through the NF-κB pathway[23,24]. The scRNA-seq data showed that CD45⁻ cells from cholesteatoma did not exhibit upregulation of IL-1β, PGE₂ synthase, or TNF-α, compared to skin fibroblasts (Supplementary Fig. S11 a–c). In addition, we reported previously that infiltrating mononuclear cells express IL-1β in human cholesteatoma[4], and previous research reported that the proportion of M1 macrophages producing inflammatory cytokines is elevated in cholesteatoma[25]. Taken together, our observations suggest that

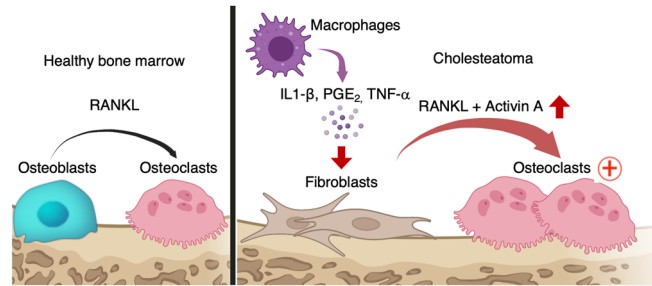

**Fig. 5 | Schematic of osteoclastogenesis induced by cholesteatoma fibroblasts expressing activin A.** IL-1β, PGE$_2$, and TNF-α secreted from infiltrating CD45$^+$ cells, particularly macrophages, induced activin A-expressing pathogenic fibroblasts; the activin A acted in conjunction with RANKL to promote ectopic osteoclastogenesis.

infiltrating CD45$^+$ cells, particularly macrophages, produce proinflammatory cytokines, thereby promoting pathogenic fibroblast differentiation and activin A secretion that results in increased ectopic osteoclastogenesis in human cholesteatoma.

We selected *TMEM119* and *PMEPA1* as possible surface markers of cholesteatoma fibroblasts in subcluster 8. Transmembrane protein 119 (TMEM119) is considered a marker for microglia[26]. A recent study showed that *TMEM119* was highly expressed in an α-SMA$^{high}$ cluster in breast cancer-associated fibroblasts; moreover, an α-SMA$^{high}$ cluster produced several growth factors involved in cancer development and progression[27]. Prostate transmembrane protein androgen induced 1 (PMEPA1) regulates cancer cell progression; it reportedly controls proton production by osteoclasts[28,29]. The expression of *TMEM119* and *PMEPA1* may reflect the characteristics of subcluster 8; these genes may promote the development and progression of bone destruction in cholesteatoma. These markers are subjects for further analysis in upcoming studies. While subcluster 9 is hypothesized as the origin of fibroblasts, additional lineage tracing and comprehensive tissue localization analysis will be required in future studies to definitively establish the source of fibroblasts.

Activin A promotes or inhibits osteoclastogenesis depending on the surrounding microenvironment and provides local regulation of bone resorption and remodeling[16,30−36]. In this study, we demonstrated that activin A-producing fibroblasts promote osteoclastogenesis in the cholesteatoma microenvironment using human scRNA-seq and a murine model of cholesteatoma. It has also been reported that activin A is involved in bone destruction in other diseases associated with inflammatory bone destruction, such as rheumatoid arthritis[34,36−38]. Fibroblast-like synoviocytes (FLS) express activin A, and disease severity is correlated with circulating activin A level[37]. Activin A from the inflammatory synovium is specifically involved in inflammatory bone/joint destruction[38]. In addition, the deletion of activin A in FLS diminishes joint pathology in experimental arthritis[36]. To explore the similarities among fibroblasts between cholesteatoma and rheumatoid arthritis, we compared data from the present study with cross-tissue, single-cell stromal atlas data[39]. We downloaded the UMI matrix and metadata from the Single Cell Portal (Broad Institute). The expression profiles of genes shared with the scRNA-seq data from the present study were normalized, clustered, and integrated using the ingest function of Scanpy. Mapping onto the cross-tissue stroma atlas data indicated that the molecular properties of fibroblasts in our study were similar to the properties of fibroblasts in the synovial membrane of joints in rheumatoid arthritis, suggesting that fibroblasts have common characteristics in cholesteatoma and rheumatoid arthritis—both constitute inflammatory bone destructive diseases (Supplementary Fig. S11d−f). Taken together, our results support the suggestion that activin A is a common key molecule for ectopic osteoclastogenesis and subsequent pathological bone destruction in inflammatory diseases.

Although we have shown that activin A is involved in ectopic osteoclast induction, activin A has also been reported to play roles in skin wound repair[40,41]. Indeed, *INHBA*-deficient mice exhibited delayed wound repair at the tympanic membrane after external trauma (Supplementary Fig. S12a, b). Our data suggest that activin A plays a role in the maintenance and wound healing of membranes and also acts as a pathogenic factor for bone destruction in cholesteatoma.

Furthermore, osteosclerosis reportedly occurs in cholesteatoma as a result of chronic inflammation[42]; activin A reportedly affects osteosclerosis in fibrodysplasia ossificans progressiva and heterotopic ossifications[43,44]. In the present study, the scRNA-seq analysis revealed the upregulation of extracellular matrix-related genes in cholesteatoma fibroblasts, including collagen proteins and proteoglycans, which are related to osteogenesis, mineral deposition, and bone remodeling[45]. These extracellular matrix genes are presumed to reflect osteogenesis and mineral deposition in cholesteatoma. The present study demonstrated a relationship between bone destruction and activin A secretion by cholesteatoma fibroblasts; these cells may be at least partially involved in osteosclerosis in cholesteatoma.

Currently, the only effective treatment for cholesteatoma is complete surgical resection; however, the rate of postoperative recurrence is high[12], and recurrence is correlated with tissue inflammation status[46]. Here, we demonstrated that the INHBA antagonist FST reduced ectopic osteoclastogenesis in a cholesteatoma mouse model. Our results suggest that activin A and the inflammatory cytokines that induce its production are potential therapeutic targets for cholesteatoma.

## Methods
### Study approval
All animal experiments were conducted following the institutional guidelines for animal experimentation, with protocols endorsed by the Animal Experimental Committee of Osaka University (FBS-19-004-6). Research involving human subjects was conducted with the approval of the Institutional Review Board at Osaka University. Appropriate informed consent was obtained from all subjects prior to their participation in the study.

### Patients and tissue samples
Fifteen cholesteatoma specimens were collected from patients who underwent tympanomastoidectomy. All specimens were obtained along with the surrounding eroded bone. We used the retroauricular skin as a control because there is no tissue in the middle ear with the same structure as cholesteatoma. Retroauricular skin was collected from the incision site. All experiments using human specimens were approved by the ethics committee of Osaka University Hospital, and informed consent was received prior to use.

### Human cell isolation
Cholesteatoma specimens and corresponding control skin specimens were cut into plugs, 4 mm in diameter, and treated using a Human Whole Skin Dissociation kit (Miltenyi Biotec, Bergisch Gladbach, Germany) according to the manufacturer's protocol and digested mechanically using a gentleMACS Dissociator (Miltenyi Biotec) with the program h_skin_01. Collected cell suspensions were filtered through 40 µm cell strainers, centrifuged at $300 \times g$ for 10 min at 4 °C, and treated with a debris removal solution kit (Miltenyi Biotec) according to the manufacturer's protocol. The obtained cells were treated with red blood cell lysis buffer (Sigma-Aldrich, St. Louis, MO, USA) and used for scRNA-seq, flow cytometry, and cell sorting.

### Flow cytometry and cell sorting
Measurements were performed using an SH 800 cell sorter (Sony Biotechnology, San Jose, CA, USA) and analyzed using FlowJo software

(TreeStar, Ashland, OR, USA). Isolated cells were treated with 0.1 µM Calcein AM solution (C396; Dojindo Molecular Technologies, Kumamoto, Japan) for 15 min at 37 °C to distinguish living cells. Next, the cells used for scRNA-seq were incubated with anti-CD45-BV421 (304031, 1:50, HI30; BioLegend, San Diego, CA, USA) antibody and 7AAD (559925, 1:50; BD Biosciences, Franklin Lakes, NJ, USA) on ice for 15 min. On the other hand, cells used for human fibroblast analysis were incubated with anti-CD45-PEcy7 (304015, 1:50, HI30; BioLegend), anti-CD31-BV421 (303123, 1:50, WM59; BioLegend) and anti-CD324-BV421 (147319, 1:50, DECMA-1; BioLegend) antibodies on ice for 15 min. Cells used for mouse fibroblast analysis were incubated with anti-CD45-PEcy7 (103113, 1:50, 30-F11; BioLegend), anti-CD31-BV421 (102423, 1:50, 390; BioLegend), and anti-CD324-BV421 (147319, 1:50, DECMA-1; BioLegend) antibodies on ice for 15 min. Fluorescence Minus One control was used as a negative control for sorting in scRNA-seq and human or mouse fibroblast analysis. Cells for scRNA-seq were added to phosphate-buffered saline (PBS), and cells for human fibroblast analysis were added directly to RLT buffer (Qiagen, Germantown, MD, USA).

### scRNA-seq analyses

Living cholesteatoma and control skin CD45⁻ cells were isolated from patients undergoing surgery. Then, targeted scRNA-seq analysis was conducted using the BD Rhapsody Single-Cell Analysis System (BD Biosciences) according to the manufacturer's protocol. Briefly, a single-cell suspension was loaded into a BD Rhapsody cartridge with >200,000 microwells, and single-cell capture was achieved by random distribution and gravity precipitation. Next, the bead library was loaded into the microwell cartridge to saturation to pair each cell in the microwell with a bead. The cells were lysed in the microwell cartridge to hybridize mRNA molecules to bar-coded capture oligos on the beads. Then, beads were collected from the microwell cartridge into a single tube for subsequent cDNA synthesis, exonuclease I digestion, and multiplex-PCR-based library construction. We used BD Rhapsody whole-transcriptome analysis for library construction. Sequencing was performed on the Illumina NovaSeq 6000 platform (Illumina, San Diego, CA, USA). The BD Rhapsody Analysis Pipeline[47] was used to process the sequencing data (FASTQ files), resulting in gene expression profiles of hashtag-attached cell barcodes. The transcriptomes of 4533 barcodes, 4501 barcodes, 3921 barcodes, 3076 barcodes, 2494 barcodes, and 748 barcodes in recursive substitution error correction reads per cell format from six different libraries of the three patients were merged and subjected to downstream analysis using Scanpy[48]. Read counts of the hashtags of the cell barcodes were extracted and scaled from 0 to 10. Then the cell barcodes originating from each sample were estimated based on the hashtag read counts of each barcode. To avoid potential bias in the clustering results, ribosome genes *TRAV*, *TRAJ*, *TRBJ*, *TRBV*, *IGHV*, *IGKV*, and *IGLV* were excluded from the downstream analysis. The resulting 19,273 cells consisting of 10,916 cells from skin specimens and 8357 cells from the cholesteatoma specimens were used in subsequent analyses. Batch effect correction was performed using Scanorama[49]. Leiden clustering[50] and PAGA graph[51] were integrated with UMAP projection[52] for whole samples or fibroblast subclusters. Cell type annotation was performed by comparing the intersection between the top DEGs in the resulting 17 clusters and the marker genes in PanglaoDB[53]. To confirm the observed bias in cell populations, the chi-square test was conducted using the SciPy Python package[54]. Trajectory mapping and detection of marker genes were conducted using Monocle 3[55] included in BioTuring BBrowser[56].

### Droplet digital PCR

RLT buffer containing human fibroblast mRNA was processed using an RNeasy kit (Qiagen) and Superscript III reverse transcriptase (Thermo Fisher Scientific, Waltham, MA, USA) to obtain cDNA for ddPCR. To quantify *INHBA*, *INHBB*, *INHA*, and *RANKL* mRNA, ddPCR was performed on the QX200 ddPCR system (Bio-Rad, Hercules, CA, USA)

using TaqMan FAM-labeled probes for *INHBA*, *INHBB*, *INHA*, and *RANKL* and HEX-labeled probes for *GAPDH* as an endogenous control (Bio-Rad). The absolute copy numbers of transcripts in the reaction samples were measured using Quanta Software version 1.7.4 (Bio-Rad).

### Double immunofluorescence staining

Double immunofluorescence staining was performed on sections 3 µm thick. After deparaffinization and rehydration, the sections were subjected to heat treatment for 10 min at 95 °C in High pH Target Retrieval Solution (Dako, Glostrup, Denmark) for antigen retrieval. Sections were incubated with anti-activin A antibody (ab56057, 1:200; Abcam, Cambridge, UK), anti-RANKL antibody (ab216484, 1:100; Abcam), anti-vimentin antibody (MAB2105-SP, 280618, 1:100; R&D Systems, Minneapolis, MN, USA), or anti-CD45 antibody (FAB3791R-025, 1:20; R&D Systems) overnight at 4 °C. After washing in PBS, the sections were incubated with Alexa Fluor 488-conjugated goat anti-rat IgG (A11006, 1:500; Invitrogen, Carlsbad, CA, USA) or Alexa Fluor 568-conjugated goat anti-rabbit IgG (A11011, 1:500; Invitrogen), as appropriate, at room temperature for 30 min. After rinsing with wash buffer, the sections were mounted on glass slides, and images were acquired by confocal microscopy (A1; Nikon, Tokyo, Japan).

### Mice

Wild-type (WT) C57BL/6J mice were purchased from Clea Japan (Tokyo, Japan). CX₃CR1-EGFP knock-in mice and TRAP promoter-tdTomato transgenic mice were previously described[18,19]. The *INHBA*^fl/fl mice, which have been described elsewhere[57], were bred with *Rosa26*^CreERT2 mice. All mice were bred and maintained under specific pathogen-free conditions. The mice involved in this study were housed under a 12 h light/dark cycle, with a stable temperature between 21.5 °C and 24.5 °C and a relative humidity range of 45–65%. They were provided with a standard laboratory chow diet and had ad libitum access to water. All of the mice used in this study were female and aged between 8 and 12 weeks. All animal experiments were performed in accordance with the institutional animal experimental guidelines, and the protocols were approved by the Animal Experimental Committee of Osaka University. Mutant mice were genotyped by PCR. To detect the *INHBA* flox allele, we used the primers 5′-ACCCACCGAAGA AGCAAAGA-3′ and 5′-GGGTCTGAGAGCCCATTGTC-3′.

### Osteoclast differentiation in vitro

Bone marrow cells derived from WT mice or CX₃CR1-EGFP/TRAP-tdTomato mice were cultured with 10 ng/mL M-CSF (R&D Systems) in α-MEM containing 10% fetal calf serum for 3 days. Then bone marrow macrophages were cultured for 3 days in the presence of 50 ng/mL RANKL (PeproTech) and 10 ng/mL M-CSF to induce differentiation into osteoclasts[58]. Human/mouse recombinant activin A (R&D Systems), mouse recombinant SFRP2 (R&D Systems), and mouse recombinant IGF2 (R&D Systems) were added at the concentrations indicated in Figure legends. Nuclei were stained with DAPI (D523; Dojindo Molecular Technologies, Kumamoto, Japan). To evaluate the inhibition of activin A-mediated osteoclast differentiation, the ALK4 inhibitor SB505124 (#S2186; Selleck Chemicals, Houston, TX, USA) was added at a concentration of 10 µM; 0.005% DMSO was used as vehicle control.

### TRAP staining and osteoclast counting

Osteoclasts derived from WT mice were subjected to TRAP staining using an Acid Phosphatase, Leukocyte (TRAP) Kit (Sigma-Aldrich). TRAP-positive multinucleated (>4 nuclei) cells were considered mature osteoclasts and counted under an optical microscope.

### Confocal imaging of cultured osteoclasts

Images of osteoclasts were acquired by confocal microscopy (A1; Nikon). The excitation laser wavelength, dichroic mirrors, and emission filters to detect EGFP were 488 nm, DM560, and BP525/50,

respectively. Those for tdTomato were 561 nm, DM640, and BP595/50, respectively, and those for DAPI were 405 nm, DM495, and BP450/50, respectively (Nikon). Images of tdTomato were binarized and measured to determine the osteoclast area using NIS Elements integrated software.

### Fibroblast isolation and culture
Specimens were obtained from the ear pinnae of WT mice, *INHBA*<sup>fl/fl</sup> mice, and *INHBA*<sup>fl/fl</sup> *Rosa26*<sup>CreERT2</sup> mice for isolation of keratinocytes and fibroblasts. Mouse-ear pinna specimens were treated with dispase (4 mg/mL) for 1 h and separated into the dermis and epidermis. The epidermis was treated with TrypLE express enzyme (Thermo Fisher Scientific) at 37 °C for 5 min, and the resulting single-cell suspensions were used as keratinocytes. The dermis was treated with dispase (4 mg/mL) and collagenase (3 mg/mL) at 37 °C for 2 h, and the obtained cells were plated in 10-cm dishes and cultured in Dulbecco's modified Eagle's medium (DMEM) (Nacalai Tesque, Kyoto, Japan) supplemented with 10% fetal bovine serum (FBS). Fibroblasts derived from *INHBA*<sup>fl/fl</sup> mice or *INHBA*<sup>fl/fl</sup>*Rosa26*<sup>CreERT2</sup> mice were cultured in DMEM for 3 days and cultured in DMEM supplemented with 800 nM 4OHT for 4 days. Fibroblasts derived from *INHBA*<sup>fl/fl</sup> mice were used as controls, and fibroblasts derived from *INHBA*<sup>fl/fl</sup> *Rosa26*<sup>CreERT2</sup> mice were used as *INHBA*-KO ear pinna-derived fibroblasts. When fibroblasts were isolated from ear pinnae and cultured as primary fibroblasts, *INHBA* was suppressed by adding 800 nM 4OHT (Supplementary Fig. S13a). Human primary skin fibroblasts (C-12302; PromoCell, Heidelberg, Germany) were cultured for 3 days in DMEM supplemented with 10% FBS containing recombinant murine IL-1β (1 ng/mL; 201-LB-005; R&D Systems), PGE₂ (10 μM; 165–10813; Wako), TNF-α (50 ng/mL; 210-TA-005; R&D Systems), or IL-6 (50 ng/mL; 201-IL-010; R&D Systems). The mass of the cholesteatoma model and mouse ear pinna specimens were dissected into plugs with a diameter of 4 mm. Then these samples were treated with dispase (4 mg/mL) and collagenase (3 mg/mL) at a temperature of 37 °C for a duration of 2 h. The cells obtained from this process were subsequently used for flow cytometry and cell sorting.

### Cholesteatoma model
Ear pinna-derived keratinocytes and fibroblasts were isolated as described above. Keratinocytes ($4.0 \times 10^6$ cells) and fibroblasts ($8.0 \times 10^5$ cells) were mixed in 100 μL of PBS and injected percutaneously into TRAP promoter-tdTomato mice under the periosteum of the calvarial bone. Mice were euthanized on day 7 after injection of keratinocytes and fibroblasts. Cholesteatoma model mice were divided into four groups: control (using *INHBA*<sup>fl/fl</sup> mouse fibroblasts), *INHBA*-KO fibroblasts (using *INHBA*<sup>fl/fl</sup>*Rosa26*<sup>CreERT2</sup> mouse fibroblasts) (Supplementary Fig. S13a), vehicle treatment (100 μL PBS), and follistatin treatment (using *INHBA*<sup>fl/fl</sup> mouse fibroblasts and follistatin treatment). For follistatin treatment, 100 μg follistatin was dissolved in 100 μL PBS and injected intraperitoneally into cholesteatoma model mice every day after injection of keratinocytes and fibroblasts.

The keratinocytes and fibroblasts were injected under the periosteum of the parietal bone of TRAP-tdTomato mice, and the parietal bone was removed 1 week later to observe the osteoclasts induced on the parietal bone surface (Supplementary Fig. S13b). After fixation with 4% paraformaldehyde (PFA) and demineralization, the parietal bone was sectioned along the periphery of the mass under a microscope, and the parietal bone was stained with Alexa Fluor 488 NHS ester. The parietal bone labeled with Alexa Fluor 488 and osteoclasts labeled with tdTomato were captured under a confocal microscope to construct whole-mount 3D images of cholesteatoma model mouse parietal bones. To detect osteoclasts induced on the parietal bone surface, the parietal bone surface was stained with Alexa Fluor 488, and the surface of osteoclasts was labeled with tdTomato. As osteoclasts inside the parietal bone were covered, those exposed on the parietal

bone surface were defined as induced osteoclasts in the cholesteatoma model (Supplementary Fig. S13c).

### Tissue clearing and staining of the parietal bone
Cholesteatoma model mice were perfused with 4% PFA, and dissected parietal bone tissues were further fixed with 4% PFA overnight at 4 °C followed by decalcification for 7 days in CUBIC-B (Tokyo Chemical Industry, Tokyo, Japan). After decalcification, samples were cut manually under a microscope along the well-defined mass lesions on the parietal bones. Samples were stained with 100 μM Alexa Fluor 488 NHS ester (A20000; Thermo Fisher Scientific) for 1 h at ambient temperature, followed by washing three times with PBS. After staining, samples were treated for 15 min at 37 °C with CUBIC-R (Tokyo Chemical Industry). After refractive index (RI) adjustment, the samples were placed in a chamber with the same RI matching solution, coverslipped, and imaged by confocal microscopy (A1; Nikon).

### Whole-mount confocal imaging of parietal bone
Images of cholesteatoma model mice were acquired by confocal microscopy (A1; Nikon). Specific parameters (excitation laser wavelength, dichroic mirrors, and emission filters) were used for the detection of Alexa Fluor 488 (488 nm, DM560, and BP525/50, respectively) and tdTomato (561 nm, DM640, and BP595/50, respectively). All specimens were observed under the same conditions, with the laser power set to 2.0, the detector sensitivity of Alexa Fluor 488 set to 20, the detector sensitivity of tdTomato set to 120, and both offsets set to 0. To obtain 3D images of the whole parietal bones, X-Y tiling images were stacked on the *Z*-axis. $X-Y$ images were acquired at $512 \times 512$ resolution with a 4.0× objective lens, and the images were tiled to include the entire parietal bone. The depth used to image the entire parietal bone was specified to capture all areas positive for Alexa Fluor 488 and tdTomato, and the image stacks were collected in vertical steps of 3 μm.

### Image data analysis
To detect osteoclasts induced on the parietal bone surface, the parietal bone surface was stained with Alexa Fluor 488, and the surface of osteoclasts was labeled with tdTomato. The threshold of binarization was set to the same value for all specimens with Imaris v.9.3.1 software (Bitplane, Concord, MA, USA). As osteoclasts inside the parietal bone were covered, those exposed on the parietal bone surface were defined as induced osteoclasts in the cholesteatoma model. TIFF images of the parietal bone surface were prepared, showing Alexa Fluor 488-labeled parietal bone and tdTomato-labeled osteoclasts. The area of osteoclasts on the parietal bone was measured using ImageJ software (NIH, Bethesda, MD, USA). The output images were divided into Alexa Fluor 488 and tdTomato channels, and the Alexa Fluor 488 images were binarized to determine the surface of the entire parietal bone. As the parietal bone was cut manually along the mass, the edges were removed by applying the erosion process 10 times in ImageJ software to remove the artificial bias of the edges, and the area of the obtained image was measured and used as the surface area of the parietal bone. After removing the background in the tdTomato channel, the tdTomato-positive area was binarized to determine the surface of osteoclasts. The area of osteoclasts induced on the parietal bone surface was measured after removing osteoclasts outside the frame of the parietal bone treated as described above. The area of osteoclasts was divided by the surface area of the parietal bone to determine the area of induced osteoclasts relative to the parietal bone area.

### Reverse transcription PCR analysis
Primary cultured fibroblasts treated with IL-1β, PGE₂, TNF-α, and IL-6 were lysed in RLT buffer (Qiagen). RT-PCR was performed using Superscript III reverse transcriptase with mRNA from primary cultured fibroblasts processed using a Qiagen RNeasy kit (Qiagen). RT-PCR was

performed for 40 cycles using a Thermal Cycler Dice Real Time System TP 800 (TaKaRa Bio, Shiga, Japan) and SYBR Premix EX Taq (TaKaRa Bio). The TaqMan gene expression assay Hs00542137_m1 (Thermo Fisher Scientific) and TaqMan Fast Advanced Master Mix (Thermo Fisher Scientific) were used to detect *PI16* expression. Gene expression was normalized relative to that of the housekeeping gene *GAPDH*, and specificity was confirmed using a dissociation curve. The relative mRNA levels of interest were calculated using the $2^{-\Delta\Delta Ct}$ method. The following specific primer pairs were used (forward and reverse, respectively): mouse *GAPDH*, 5′-TGTGTCCGTCGTGGATCTGA-3′ and 5′-CCTGCTTCACCACCTTCTTGAT-3′; mouse *INHBA*, 5′-GGAGAACGG GTATGTGGAGA-3′ and 5′-TGGTCCTGGTTCTGTTAGCC-3′; mouse *INHBB* 5′-AGGCAACAGTTCTTCATCGACTTTC-3′ and 5′-AGCCACACTC CTCCACAATCATG-3′; mouse *INHA* 5′-TTCATTTTCCACTACTGCCA TGGTA-3′ and 5′-GATACAAGCACAGTGTTGTGTAATG-3′; human *GAPDH* 5′-CATGTTCGTCATGGGTGTGA-3′ and 5′-CATGAGTCCTTC CACGATACCA-3′; and human *INHBA* 5′-TCATGCCAACTACTGCGAGG-3′ and 5′-ACAGTGAGGACCCGGACG-3′.

## Mouse model of tympanic membrane perforation

Mice with tympanic membrane perforation were divided into control (*INHBA*^fl/fl mice) and *INHBA*-KO (*INHBA*^fl/fl*Rosa26*^CreERT2 mice) groups. Tympanic membrane perforation was performed in the bilateral ears of all mice by puncture using a 27 G needle under microscopic visualization. All mice were euthanized on day 3 after tympanic membrane perforation. We planned to administer pain relief if mice showed signs of suffering. However, no drugs were used because the animals showed no signs of suffering. The tympanic membranes were visualized under microscopy, and the size of the hole was measured using ImageJ.

## Statistical analysis

Data were analyzed using GraphPad Prism ver. 6 (GraphPad Software, San Diego, CA, USA). Data are presented as the mean ± SD unless otherwise stated. Statistical analyses were performed using the two-tailed Student's t-test for in vitro and in vivo analyses, ratio paired t-test for comparison of human specimens between two groups, and one-way ANOVA with Tukey's post hoc multiple comparison test for comparisons among three or more groups. In all analyses, $P < 0.05$ was taken to indicate statistical significance. Statistical significance in marker genes reflecting pseudotime trajectory was determined using the Moran's I test. The enrichment analysis in subcluster 8 was conducted through gprofiler[59] integrated with scanpy v1.9, as well as DAVID[60] (accessed on 7/8, 2021). The data shown are representative of at least three independent experiments unless otherwise indicated. We estimated the sample size considering the variation and mean of the sample and attempted to reach a conclusion using as small sample size as possible. Sex- and age-matched mice were randomly assigned to groups for in vivo experiments, and no data points were excluded. Investigators were not blinded during the experiments or outcome assessment.

## Statistics and reproducibility

We did not employ any statistical method to predetermine the sample size. Sample sizes were estimated based on considering variation and means, aiming to achieve reliable conclusions with the smallest possible number of samples. We considered previously published results, experimental complexity, cost, and past experience to determine sample size. No data were excluded from our analysis.

Sex was not considered in the study design due to insufficient sample size to analyze sex differences. This study involves cholesteatoma samples and retroauricular skin from the incision site of cholesteatoma patients. All fifteen cholesteatoma and skin specimens were gathered from patients undergoing tympanomastoidectomy. Patients were not compensated for their participation.

Patients who had a sufficient sample volume available for experimental analysis were selected for the study. However, the potential bias of excluding patients with mild symptoms could possibly skew the study's results, which makes them appear more severe compared to the overall average of all cholesteatoma patients.

Experiments were conducted with a sufficient sample size to ensure the reproducibility of the findings. All experiments reported in this study were independently repeated at least three times, with all replication attempts being successful.

For in vivo studies, mice were randomly assigned to each treatment group within each genotype. For in vitro studies, conditions were randomly assigned to each experimental condition. Blinding was deemed irrelevant for all assays, as the same researcher was involved in all experimental procedures and analyses.

## Reporting summary

Further information on research design is available in the Nature Portfolio Reporting Summary linked to this article.

## Data availability

The single-cell RNA sequencing data have been deposited in the NCBI Gene Expression Omnibus (GEO) database with the accession number GSE210261 Marker gene data for cell type annotation can be found at PanglaoDB53 [https://panglaodb.se/index.html]. All other data that support the findings of this study are available within the article and its supplementary information files or from the corresponding authors upon reasonable request. Source data are provided in this paper.

## Code availability

All source code has been made publicly available via Zenodo[61].

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

## Acknowledgements

We would like to thank Drs. Erika Yamashita, Tetsuo Hasegawa, and Seiji Taniguchi for their useful discussions and Mrs. Ayumi Sakai and Maki Kawasaki for their technical assistance. This work was supported by CREST (to M.I.), Japan Science and Technology (JST) Agency; Grant-in-Aid for Scientific Research (to M.I., J.K., and K.O.), Grant-in-Aid for Transformative Research Areas (to J.K.), and Grant-in-Aid for Young Scientists and Research Activity Start-up (to K.S.) from the Japan Society for the Promotion of Science (JSPS) Grant Numbers 19H01044, 19H05657, 19K09844, 22H05083, 19K23826, 20K18312, and 21H02716; funding from PRIME, Japan Agency for Medical Research and Development (to J.K.); grants from the Uehara Memorial Foundation (to M.I.), the Kanae Foundation for the Promotion of Medical Sciences (to M.I.), the Mochida Memorial Foundation (to M.I.), and the Takeda Science Foundation (to M.I. and J.K.). Illustrations in Figs. 4c and 5 were created with BioRender.com.

## Author contributions

K.S., H.I., and M.I. conceived the study. K.S. performed RNA-seq analysis with the assistance of Y.L., D.O., and D.M. K.S., Y.U., A.M., Y.M., and S.Y. performed the in vitro and in vivo experiments with the assistance of J.K. and R.I. Y.O. performed the surgery for collection of human samples with T.S., T.K., and K.O. N.E. generated the INHBAfl/fl mouse line with T.H. K.S. wrote the initial draft. J.K. and M.I. revised the final draft.

## Competing interests

The authors declare no competing interests.
