## [Peer Review File · Nature Communications]

Single-cell transcriptomics of human cholesteatoma identifies an activin A-producing osteoclastogenic fibroblast subset inducing bone destructionREVIEWER COMMENTS

Reviewer #1 (Remarks to the Author):

Shimizu et al. performed single-cell RNA-seq to study the pathological mechanisms in human cholesteatoma. The authors thereby identified a pathogenic fibroblast population causing bone destruction by secreting inhibin β A which stimulates osteoclast differentiation. Thus, this study reveals the previously unknown mechanisms underlying cholesteatoma-associated bone destruction and propose inhibin β A-producing fibroblasts as therapeutic target in human cholesteatoma.

Major comments:

1.) The authors identify a disease-associated fibroblast population that is not patient-dependent and differs from all other fibroblast populations. Would it be possible to show a Gene Ontology (GO) and/or a Gene Set Enrichment Analysis (GSEA) to show differences in the cell signaling pathways of this particular fibroblast subpopulation? Can the author mine the transcriptomics data and comment on possible surface markers to identify and maybe isolate these pathogenic fibroblasts specifically? Suppl. Figure S3 does not seem to discriminate between the different fibroblast populations.

2.) Except for INHBA the authors do not investigate any other gene, also not the other 2 secreted genes SFRP2 and IGF2. Can the authors comment, whether these are relevant for cholesteatoma biology or not. How do the authors explain the upregulation of extracellular matrix-associated genes in the cholesteatoma-specific subset?

3.) The authors identify inhibin β A-mediated osteoclast differentiation based on the TRAP-marker gene expression of activating A/RANKL-treated macrophages isolated from mice. However, osteoclasts and macrophages are derived from competing lineages from myeloid progenitors. Is it possible to show the cells before and after differentiation in high resolution and provide an immunofluorescence analysis for one or two more characteristic markers for macrophages (before) and osteoclasts (after). This would help to strengthen the main point being osteoclast differentiation. Figure S7 is not sufficiently evidencing the differentiation, especially panel b does not well visualize the proportion of osteoclasts being formed.

Reviewer #2 (Remarks to the Author):

In this manuscript, Shimizu et al. utilized single-cell RNAseq (scRNAeq) to examine the cellular components of cholesteatoma. The authors identified a fibroblast population expanded in cholesteatoma. These fibroblasts express high levels of INHBA. In vitro experiments suggest that Activin A did not induce the formation of osteoclasts cells but could promote the induction effect while in concert with RANKL. The authors also provided in vivo data using a mouse model of cholesteatoma which suggests that the INHBA-deleted fibroblasts group resulted in fewer osteoclasts formation on the parietal bone surface, compared with the control group.

The data is presented in this manuscript. However, activin A has already been reported to enhance RANKL-induced osteoclast, which limits the novelty of this study. Secondly, further direct, functional experiments are needed to support the mechanism suggested by the authors. Lastly, the analysis of single-cell RNAseq is extremely limited in terms of sample number.

1. In Figure 1b, there appears to be very little mixing of cells derived from different experimental conditions. Did the authors consider applying data integration methods, such as Harmony, to correct for experimental batch effect in single-cell RNAseq data?

2. The study is limited by the number of patient samples (n=3). In Figure 2, it is not clear to

this reviewer whether the differences in cell type abundance between disease and healthy skin are statistically significant. The proportion of each cell type and cell state should be quantified, and statistical differences between the disease state and healthy state should be determined.

3. The authors utilized pseudotime trajectory analysis to infer fibroblast differentiation, specifically implicating that “cholesteatoma-specific fibroblasts differentiated from subcluster 2 toward subcluster 8, via subclusters 10, 7, and 11.” Trajectory analysis of single-cell RNAseq data suggests a transcriptional relationship between cell states. The author should provide experimental data, in vitro or in vivo, to show that these states indeed represent fibroblast differentiation states and that the transcriptional state of subcluster 8 can be derived from subcluster 2.

4. The author suggests that “upregulated INHBA in cholesteatoma-specific fibroblasts forms a homodimer, activin A” based on the lack of detection of INHBB and INHA from the single-cell RNAseq data. Since single-cell RNAseq data only detect highly expressed genes in individual cells, the lack of detection in scRNAseq data does not exclude a lower level of gene expression that fails to be captured using this technique. The authors need alternative methods to evaluate and differentiate between the levels of activin A, activin AB, and inhibin A in cholesteatoma-specific fibroblasts.

5. Fig 4c. In the transplantation experiments of Fig4c, the INHBA KO group resulted in fewer osteoclasts. Therefore, the authors suggest that INHBA/activin A from the pathogenic fibroblast was a key molecule for ectopic osteoclastogenesis. However, these experimental results can also be explained in other ways. For example, the INHBA knockout can lead to the depletion of the fibroblast after transplantation. Alternatively, INHBA may form activin AB or inhibin A which make contribute to these results, since the author did not show the expression level of INHBB and INHA in mouse fibroblast. Additional experiments are necessary to be shown. Such as the pattern of transplanted fibroblast (which may form epidermal cysts according to the authors’ previous work (Yoriko Iwamoto, 2015)). The expression level of INHBB and INHA in mouse fibroblast needs to be detected. The author also needs to check the RANKL expression level in the cultured fibroblasts that are ready to be injected.

6. Fig4d. The positive control was provided as Control/FST- group. A negative control, such as the PBS injection group, is necessary for a better understanding of these results.

Reviewer #3 (Remarks to the Author):

This is a very impressive study that provides a cellular and molecular mechanism to explain why bone erosions occur in cholesteatoma. Using scRNA analysis of samples from four patients with cholesteatoma and four control samples the authors identify enrichment of a pathogenic subset of fibroblasts that express inhibinBA and with RANKL drive osteoclastogenic differentiation. Using both genetic and pharmacological inhibition of inhibinBA in fibroblasts they show that bone erosions are reduced in a mouse model of cholesteatoma.

While the authors have done a first-rate job in revealing a new and exciting mechanism, I have a few issues with the interpretation of their data which I hope, if addressed, would make their findings even more interesting

Major points

1. The nature of the control tissue (retro auricular skin) is important as all further comparisons of enrichment of populations in the cholesteatoma are made against this control. Skin is derived from ectoderm whereas the embryology of the middle ear requires many separate components from different embryonic origins and in particular form from the mesoderm of the branchial pouches. I am concerned that the comparison of cholesteatoma fibroblasts to skin fibroblast is not a fair one and a better one would be with mesenchymal fibroblast from for example synovial tissue. The auditory ossicles are connected by synovial

joints so this would be a better control even if taken from a large synovial joint. What would happen to the analysis if the data in Fig 2c were a comparison against publicly available data sets of (for example PMID: 35649411 or PMID: 31061532) synovial fibroblasts compared to their retro auricular skin skin (control) fibroblasts.

2. I do not understand why the authors use cluster 2 as the common origin for cholesteatomata and control (skin) fibroblasts. Does this cluster contain Pi16 the recently described marker for universal fibroblasts (PMID: 33981032). If it does then I could understand the choice of cluster 2 for the pseudotime trajectory. If not, it seems a rather random choice to start the trajectory analysis from cluster 2. What other genes in subset 2 make the authors think this is a common origin fibroblast?

Minor Points

3. Why are mouse ear pinna derived fibroblasts used in Fig 3g and not human retro auricular skin fibroblasts

4. In the discussion (line 245-248) the authors mention that infiltrating CD45+ cells produce proinflammatory cytokines and PGE2. Do they think these are macrophages and/ or granulocytes? If so could these cells be added to Fig 5 as the source of these factors

Response to the Reviewers

We thank the reviewers for the positive and constructive comments about our manuscript, **“Single-cell transcriptomics of human cholesteatoma identifies an activin A-producing osteoclastogenic fibroblast subset inducing bone destruction”** (NCOMMS-22-47016), by Shimizu *et al.* We performed additional experiments and revised the manuscript in response to the comments. We hope that we have responded adequately and that the revised manuscript is now suitable for publication.

Response to Reviewer #1

[Comment]

Shimizu et al. performed single-cell RNA-seq to study the pathological mechanisms in human cholesteatoma. The authors thereby identified a pathogenic fibroblast population causing bone destruction by secreting inhibin β A which stimulates osteoclast differentiation. Thus, this study reveals the previously unknown mechanisms underlying cholesteatoma-associated bone destruction and propose inhibin β A-producing fibroblasts as therapeutic target in human cholesteatoma.

Major comments.

1.) The authors identify a disease-associated fibroblast population that is not patient-dependent and differs from all other fibroblast populations. Would it be possible to show a Gene Ontology (GO) and/or a Gene Set Enrichment Analysis (GSEA) to show differences in the cell signaling pathways of this particular fibroblast subpopulation? Can the author mine the transcriptomics data and comment on

possible surface markers to identify and maybe isolate these pathogenic fibroblasts specifically? Suppl. Figure S3 does not seem to discriminate between the different fibroblast populations.

[Response]

Thank you for the instructive comment. As the reviewer suggested, we performed Gene Ontology (GO) analysis and picked up the biological pathways upregulated in subcluster 8. The results showed that subcluster 8 was correlated with multicellular organism development, unlike other cholesteatoma-specific fibroblast subpopulations (**revised Supplementary Fig. S5d**). Multicellular organism development (GO: 0007275) was defined as “the biological process whose specific outcome is the progression of a multicellular organism over time from an initial condition (*e.g.*, a zygote or a young adult) to a later condition (*e.g.*, a multicellular animal or an aged adult)” in the GO term, suggesting that subcluster 8 was a differentiated state of fibroblasts. This was consistent with our trajectory mapping, which showed that subcluster 8 constituted the most differentiated fibroblasts in cholesteatoma (**Fig. 3b**).

We also identified the cell surface markers for cholesteatoma-specific fibroblast subcluster 8. We investigated the top 100 genes upregulated in subcluster 8, and identified the cell surface and transmembrane proteins. We focused on *TMEM119* and *PMEPA1* as candidate surface markers (**revised Supplementary Fig. S6**). *TMEM119* (transmembrane protein 119) is a known microglial marker (Honarpisheh *et al.*, *J Neuroinflammation*, 2020). It has recently been reported that *TMEM119* is highly expressed in α -SMA^{high} cluster breast cancer-associated fibroblasts, and that the α -SMA^{high} cluster produces several growth factors involved in cancer development and progression (Sebastian *et al.*, *Cancers*, 2020). Meanwhile, *PMEPA1* (prostate transmembrane protein androgen

induced 1) regulates cancer cell progression (Rae *et al.*, *Mol Carcinog*, 2001). It has been reported that *PMEPA1* controls proton production in osteoclasts (Xu, *FASEB J.*, 2019). It is possible that *TMEM119* and *PMEPA1* reflect the characteristics of subcluster 8 and promote development, progression, and bone destruction in cholesteatoma. *TMEM119* and *PMEPA1* are highly upregulated in subcluster 8, although other cholesteatoma fibroblasts, such as subcluster 11, also express *TMEM119* and *PMEPA1* (**revised Supplementary Fig. S6**). In response to the reviewer's comments, we identified the candidate cell surface markers. Nevertheless, the isolation of *TMEM119*⁺ and *PMEPA1*⁺ fibroblasts was limited because of the lack of access to human specimens. These markers will be further analyzed in future studies.

We added the data in **revised Supplementary Fig. S5d, S6** and revised the text as follows;

*“This was consistent with the results of Gene Ontology (GO) biological pathway analysis, which showed that subcluster 8 was correlated with multicellular organism development relative to other cholesteatoma-specific fibroblast subpopulations (Supplementary Fig. S5d). We also identified cell-surface markers of cholesteatoma-specific fibroblast subcluster 8. We investigated the top 100 genes upregulated in subcluster 8, then selected cell surface or transmembrane proteins. Among these genes, we identified *TMEM119* and *PMEPA1* as candidate surface markers (Supplementary Fig. S6).”* (page 11, line 3-10)

*“We selected *TMEM119* and *PMEPA1* as possible surface markers of cholesteatoma fibroblasts in subcluster 8. *TMEM119* (transmembrane protein 119) is regarded as a marker for microglia²⁶. A recent study showed that *TMEM119* was highly expressed in an α -SMA^{high} cluster in breast cancer-associated fibroblasts; moreover, an α -SMA^{high} cluster produced several growth factors involved in cancer development and progression²⁷.*

PMEP1 (prostate transmembrane protein androgen induced 1) regulates cancer cell progression; it reportedly controls proton production by osteoclasts^{28,29}. The expression of TMEM119 and PMEP1 may reflect the characteristics of subcluster 8; these genes may promote the development and progression of bone destruction in cholesteatoma. These markers will be further analyzed in future studies.” (page 19, line 13- page 20, line 6)

[Comment]

2.) Except for INHBA the authors do not investigate any other gene, also not the other 2 secreted genes SFRP2 and IGF2. Can the authors comment, whether these are relevant for cholesteatoma biology or not. How do the authors explain the upregulation of extracellular matrix-associated genes in the cholesteatoma-specific subset?

[Response]

We appreciate the comment. In the present study, we investigated whether two of the other secreted proteins, such as SFRP2 and IGF2, could promote osteoclast differentiation. Bone marrow cells derived from wild-type mice were cultured with M-CSF for 3 days. These cells were further cultured for 3 days in the presence of RANKL and M-CSF with human recombinant SFRP2 or IGF2, followed by TRAP staining. As shown in the **revised Supplementary Fig. S8b–e**, SFRP2 and IGF2 did not promote osteoclast differentiation. Therefore, we focused on activin A as the osteoclast forming protein in cholesteatoma.

As pointed out by the reviewer, it was necessary to explain the upregulation of extracellular matrix-associated genes in the cholesteatoma-specific fibroblast subset. According to a previous study, chronic inflammation causes osteosclerosis in

cholesteatomas (Martin *et al.*, *Acta Otolaryngol*, 2006). It has been also reported that activin A affects osteosclerosis in fibrodysplasia ossificans progressiva and heterotopic ossifications (Hino *et al.*, *Proc Natl Acad Sci USA*, 2015; Mundy *et al.*, *Sci Signal*, 2021). In this study, scRNA-seq results showed upregulation of extracellular matrix-related genes in cholesteatoma fibroblasts. This indicates a relationship between bone destruction and activin A secretion by cholesteatoma fibroblasts; it is possible that cholesteatoma fibroblasts are at least partly involved in osteosclerosis of cholesteatomas.

We added the data in **revised Supplementary Fig. S8b–e** and revised the text as follows;

“SFRP2 and IGF2 (secreted proteins that are upregulated in cholesteatoma fibroblasts except activin A) did not promote osteoclast differentiation; therefore, we used activin A protein for further in vitro experiments (Supplementary Fig. S8b–e).” (page 14, line 10-13)

“Furthermore, osteosclerosis reportedly occurs in cholesteatoma as a result of chronic inflammation⁴²; activin A reportedly affects osteosclerosis in fibrodysplasia ossificans progressiva and heterotopic ossifications^{43, 44}. In the present study, scRNA-seq analysis revealed upregulation of extracellular matrix-related genes in cholesteatoma fibroblasts, including collagen proteins and proteoglycans, which are related to osteogenesis, mineral deposition, and bone remodeling⁴⁵. These extracellular matrix genes are presumed to reflect osteogenesis and mineral deposition in cholesteatoma. The present study demonstrated a relationship between bone destruction and activin A secretion by cholesteatoma fibroblasts; these cells may be at least partially involved in osteosclerosis in cholesteatoma.” (page 22, line 4-13)

[Comment]

3.) The authors identify inhibin β A-mediated osteoclast differentiation based on the TRAP-marker gene expression of activating A/RANKL-treated macrophages isolated from mice. However, osteoclasts and macrophages are derived from competing lineages from myeloid progenitors. Is it possible to show the cells before and after

differentiation in high resolution and provide an immunofluorescence analysis for one or two more characteristic markers for macrophages (before) and osteoclasts (after). This would help to strengthen the main point being osteoclast differentiation. Figure S7 is not sufficiently evidencing the differentiation, especially panel b does not well visualize the proportion of osteoclasts being formed.

[Response]

We appreciate the reviewer's comments. To obtain high resolution images, we performed an osteoclast differentiation assay with double-reporter mice; CX₃CR1-EGFP knock-in mice (Jung *et al.*, *Mol Cell Biol*, 2000) and TRAP promoter-dependent tdTomato-expressing (TRAP-tdTomato) mice (Kikuta *et al.*, *J Clin Invest*, 2013) instead of immunofluorescence staining. CX₃CR1 is a fractalkine receptor and a characteristic marker for macrophages, while TRAP is a marker for mature osteoclasts. Bone marrow-derived macrophages from CX₃CR1-EGFP/TRAP-tdTomato mice were cultured in the presence or absence of activin A and/or RANKL, and the area of TRAP⁺ osteoclasts was quantified. Activin A did not induce TRAP⁺ mature osteoclasts, but the combination of activin A and RANKL synergistically increased the number of TRAP⁺ mature osteoclasts (**revised Fig. 4a, b**). In addition, mature osteoclasts decreased the CX₃CR1 expression, and the number of EGFP⁺ cells was decreased after osteoclast differentiation (**revised Supplementary Fig. S9c**). The images before the differentiation assay were similar to the images without RANKL/Activin A stimulation, and there were no detectable tdTomato⁺ cells. We also revised the images showing the effects of ALK-4 inhibitor using bone marrow-derived macrophages from double reporter mice, and obtained effects similar to those reported previously (**revised Supplementary Fig. S9d, e**). Furthermore, we performed the *in vitro* experiments for osteoclast differentiation and acquired high

resolution images (**revised Supplementary Fig. S9a, b**).

We revised the data in **revised Fig. 4a, b, Supplementary Fig. S9** and rewrote the text as follows;

*“To evaluate the osteoclast differentiation, we used double-reporter CX₃CR1-enhanced green fluorescent protein (EGFP) knock-in (CX₃CR1-EGFP) mice¹⁸ and tartrate-resistant acid phosphatase (TRAP) promoter-dependent tdTomato-expressing (TRAP-tdTomato) mice¹⁹. CX₃CR1, a fractalkine receptor, is regarded as a marker of monocyte-lineage cells, including osteoclast precursors; TRAP is a marker of mature osteoclasts. Bone marrow-derived macrophages derived from CX₃CR1-EGFP/TRAP-tdTomato mice were cultured in the presence or absence of activin A and/or RANKL; subsequently, the area of TRAP⁺ osteoclasts was quantified. We found that activin A did not induce TRAP⁺ mature osteoclasts, whereas the combination of activin A with RANKL synergistically increased the number of mature TRAP⁺ osteoclasts (**Fig. 4a, b, Supplementary Fig. S9a–c**). An inhibitor of the activin A receptor ALK-4 suppressed the synergistic effect of activin A, as well as the basal effect of RANKL on osteoclastogenesis (**Supplementary Fig. S9d, e**), suggesting that activin A is a key cofactor for RANKL-dependent osteoclastogenesis.” (page 14, line 16- page 15, line 13)*

Response to Reviewer #2

[Comment]

In this manuscript, Shimizu et al. utilized single-cell RNAseq (scRNAeq) to examine the cellular components of cholesteatoma. The authors identified a fibroblast population expanded in cholesteatoma. These fibroblasts express high levels of INHBA. In vitro experiments suggest that Activin A did not induce the formation of osteoclasts cells but could promote the induction effect while in concert with RANKL. The authors also provided in vivo data using a mouse model of cholesteatoma which suggests that the INHBA-deleted fibroblasts group resulted in fewer osteoclasts formation on the parietal bone surface, compared with the control group.

The data is presented in this manuscript. However, activin A has already been reported to enhance RANKL-induced osteoclast, which limits the novelty of this study. Secondly, further direct, functional experiments are needed to support the mechanism suggested by the authors. Lastly, the analysis of single-cell RNAseq is extremely limited in terms of sample number.

[Response]

We greatly appreciate the insightful comments that have helped us to improve our manuscript. We understand the reviewer's concerns about novelty, mechanism, and sample number. We performed several additional experiments, including RNA-seq data analyses, that further support the validity and novelty of our original conclusion that an activin A-producing osteoclastogenic fibroblast subset induces bone destruction. We would like to separately address the three points raised by the reviewer.

First, in terms of novelty, it has indeed been previously reported that activin A

promotes RANKL-induced osteoclast differentiation. However, the mechanisms underlying local bone destruction in cholesteatoma patients remain unknown. Many hypotheses have been suggested, including osteoclast activation. Considering this controversy, we conducted single-cell RNA-seq analysis of specimens from cholesteatoma patients. To our knowledge, we are the first group to perform such an experiment. Here, we identified a specific INHBA-expressing fibroblast subset that induces local bone destruction through osteoclast activation. Our single-cell RNA-seq analysis and the findings of subsequent investigations strongly support the osteoclast activation hypothesis, potentially resolving the unknown mechanism. Additionally, studies regarding activin A-producing cell types are limited, and the *in vivo* microenvironment related to activin A-associated osteoclastogenesis in bone destructive diseases remains elusive. We identified a specific, human fibroblast subset that expresses a high level of INHBA; such cells may serve as a therapeutic target. We believe that this finding is both significant and novel.

Second, regarding the mechanism, we suggest that normal fibroblasts exposed to inflammatory signals differentiate into pathological fibroblasts expressing INHBA, which then stimulate osteoclasts to undergo osteoclastogenesis; this process results in local bone destruction. We experimentally showed that normal fibroblasts stimulated with inflammatory mediators (e.g., IL1- β , PGE₂, and TNF- α) expressed *INHBA*; moreover, the level of the universal fibroblast marker *PII6* decreased (**revised Supplementary Fig. S7d**). We believe that this experiment directly reveals the mechanism by which pathological fibroblasts are induced. The finding that disease-associated fibroblasts can induce osteoclast activity was validated in our *in vivo* cholesteatoma model. Analysis of INHBA-knockout fibroblasts in this model showed that osteoclastogenesis was

dramatically suppressed, suggesting that fibroblast-derived *INHBA* activates osteoclastogenesis in cholesteatoma patients. We believe that the additional experimental results in the revised manuscript clearly reveal the molecular mechanism involved.

Finally, in terms of sample size, we agree that this study included a small number of patients. This sample size results from disease development in the middle ear, which causes cholesteatoma masses to generally be small, thereby limiting the number of patients with masses sufficiently large for precise single-cell RNA-seq analysis. To compensate for this, we performed ddPCR analysis (**Fig. 3f; n = 6**) and additional RNA-seq analysis of the perimatrices of cholesteatomas sectioned via laser microdissection. We found that *INHBA* expression was significantly elevated in the cholesteatoma perimatrix, compared with the dermis (**Reviewer-only Fig. 1; n = 6**) ($p = 0.0043$, fold change = 3.131). These RNA-seq data support the conclusion that the perimatrix, including the fibroblasts within, of cholesteatoma expresses *INHBA* at a high level. Although our conclusion (i.e., *INHBA* from disease-associated fibroblasts induced osteoclastogenesis) was initially derived via single-cell RNA-seq analysis of a small number of samples, we have now confirmed this finding in cholesteatoma using several alternative methods with adequate sample sizes. We believe that the additional data overcome the small sample size involved in the single-cell RNA-seq analysis.

Reviewer-only Figure 1

We have performed several additional experiments for a more in-depth examination of the molecular pathophysiology of cholesteatoma. We hope that we have sufficiently addressed the reviewer's concerns.

[Comment]

1. In Figure 1b, there appears to be very little mixing of cells derived from different experimental conditions. Did the authors consider applying data integration methods, such as Harmony, to correct for experimental batch effect in single-cell RNAseq data?

[Response]

We thank the reviewer for this important comment on the batch effect correction. The batch effect of the single-cell RNA-seq analysis was corrected through *Scanorama*, as described in the manuscript (Hie *et al.*, *Nat Biotechnol*, 2019). The heterogeneous properties of different samples were retained using the applied algorithm described in the Hie *et al.* study. Although several batch effect correction algorithms have been reported, benchmarking was beyond the scope of this study.

[Comment]

2. The study is limited by the number of patient samples (n=3). In Figure 2, it is not clear to this reviewer whether the differences in cell type abundance between disease and healthy skin are statistically significant. The proportion of each cell type and cell state should be quantified, and statistical differences between the disease state and healthy state should be determined.

[Response]

We fully understand the reviewer's point and apologize for the confusion caused by the statistical analysis. We evaluated the proportion difference of disease and healthy states (**Fig. 2a**), and the sample bias (**Fig. 2b**) for each cluster using Chi-square test. A summary of the statistical analyses has been included in the **Supplemental Figure S4**. As we had obtained 8,357 cholesteatoma cells and 10,916 control skin cells, the expected disease-to-health ratio was $8,357/10,916 = 0.766$. We quantified the disease-to-health ratio for each cell cluster and then compared the values with the expected value of 0.766. If the cluster was in a diseased state, the ratio would exceed the expected value. The chi-square test was used to assess whether the difference from the expected value was statistically significant. The bias between the samples was also evaluated using the same method. Thus, we found that cluster 5 was biased toward the diseased state and unbiased among the patients.

We have also performed a condition-wise quantification of the cell proportions (**Reviewer-only Fig. 2**). The proportions of cholesteatoma and skin cells within each type are shown in **Reviewer-only Fig. 2a**. The proportions of each cell type among the cholesteatoma and skin cell groups are shown in **Reviewer-only Fig. 2b**. As shown in **Fig. 2c**, cholesteatomas and skin are mainly composed of keratinocytes, endothelial cells, and fibroblasts. The proportions of the cell types within each patient are shown in **Reviewer-only Fig. 2c**. The proportion of fibroblasts between cholesteatomas and skin did not show any statistically significant difference (**Reviewer-only Fig. 2d**) ($p = 0.3556$). Our findings suggest that the qualitative differences among cell types are more important than the proportion of cells.

Reviewer-only Figure 2

[Comment]

3. The authors utilized pseudotime trajectory analysis to infer fibroblast differentiation, specifically implicating that “cholesteatoma-specific fibroblasts differentiated from subcluster 2 toward subcluster 8, via subclusters 10, 7, and 11.” Trajectory analysis of single-cell RNAseq data suggests a transcriptional relationship between cell states. The author should provide experimental data, in vitro or in vivo, to show that these states indeed represent fibroblast differentiation states and that the transcriptional state of subcluster 8 can be derived from subcluster 2.

[Response]

We agree that the trajectory analysis for *in vitro* cell differentiation is required. We identified the marker genes in the pseudotime trajectory analysis using *Monocle 3* and focused on the universal fibroblast marker gene *PII6* (Buechler *et al.*, *Nature*, 2021). Since *PII6* was most highly expressed in subcluster 9, i.e., the root of the pseudotime trajectory, and downregulated in subcluster 8, i.e., the end of the pseudotime trajectory, the trajectory analysis suggested that subcluster 8 (cholesteatoma-specific fibroblast) differentiated from subcluster 9 via subclusters 3, 2, 10, 7, and 11 (**revised Supplementary Fig. S5a, b; revised Fig. 3b**). Therefore, *PII6* was considered a possible marker gene for tracing the trajectory.

We conducted *in vitro* experiments to confirm *PII6* gene downregulation with IL1- β , PGE₂, TNF- α , and IL-6 stimulation to promote the differentiation of *INHBA*-expressing cholesteatoma-specific fibroblasts. We used human skin fibroblasts instead of mouse fibroblasts, and cultured them with the inflammatory cytokines. IL1- β , PGE₂, and TNF- α significantly decreased the expression of the universal fibroblast marker gene, *PII6* (**revised Supplementary Fig. S7d**), and increased the expression of the subcluster 8 marker gene, *INHBA* (**Fig 3g**). These *in vitro* experiments demonstrated successful reconstitution of the trajectory, i.e., *PII6* downregulation and *INHBA* upregulation, indicating that the *INHBA*-expressing cholesteatoma-specific fibroblasts differentiated from normal fibroblasts upon inflammatory cytokine stimulation.

We added the data in **revised Supplementary Fig. S5a, b, S7d** and revised the text as follows;

“To infer the process of development from normal to disease-specific fibroblasts, we classified each fibroblast cluster into smaller subclusters and performed pseudotime trajectory analysis using Monocle 3 (Fig. 3a, b). Subcluster analysis revealed that the

fibroblasts contained 15 subclusters. Cholesteatoma-specific fibroblasts were divided into five subclusters designated as 1, 7, 8, 10, and 11 (**Fig. 3a**). We identified the top 10 marker genes indicating the differentiation status of fibroblasts, then selected the PII6 gene (described as a universal fibroblast marker in a previous study¹⁵) (**Supplementary Fig. S5a**). Subcluster 9 was defined as the origin of fibroblasts for pseudotime trajectory analysis because cells in this subcluster showed high levels of PII6 expression (**Supplementary Fig. S5b**). Pseudotime trajectory analysis suggested that the cholesteatoma-specific fibroblasts differentiated from subcluster 9 toward subcluster 8, via subclusters 3, 2, 10, 7, and 11 (**Fig. 3b**). The marker gene PII6 was downregulated after differentiation (**Supplementary Fig. S5b**). Subcluster 8 was composed of the most differentiated cholesteatoma-specific fibroblasts and exhibited markedly higher *INHBA* expression level than the other subclusters in the cholesteatoma-specific fibroblasts (**Fig. 3b–d, Supplementary Fig. S5c**).” (page 10, line 4- page 11, line 3)

“Among the proinflammatory cytokines, *IL-1 β* , *PGE₂*, and *TNF- α* significantly promoted *INHBA* expression in human primary skin fibroblasts, suggesting the significance of proinflammatory cytokine signals for *INHBA* expression and pathological fibroblast differentiation (**Fig. 3g**). Additionally, universal fibroblast marker gene PII6 mRNA expression was decreased by proinflammatory cytokines (**Supplementary Fig. S7d**), suggesting that inflammatory cytokines cause differentiation from human skin fibroblasts with general characteristics to cholesteatoma-specific fibroblasts expressing high levels of *INHBA*.” (page 13, line 7-14)

[Comment]

4. The author suggests that “upregulated *INHBA* in cholesteatoma-specific fibroblasts forms a homodimer, activin A” based on the lack of detection of *INHBB* and *INHA* from the single-cell RNAseq data. Since single-cell RNAseq data only detect highly expressed genes in individual cells, the lack of detection in scRNAseq data does not exclude a lower level of gene expression that fails to be captured using this technique. The authors need alternative methods to evaluate and differentiate between the levels of activin A, activin AB, and inhibin A in cholesteatoma-specific fibroblasts.

[Response]

In response to the reviewer's suggestion, we performed additional experiments to compare *INHBB* and *INHA* expressions in cholesteatoma fibroblasts in addition to the ddPCR analysis of *INHBA*. The results showed that *INHBB* and *INHA* expression in human cholesteatoma fibroblasts were significantly lower than the *INHBA* expression in human cholesteatoma fibroblasts (revised Supplementary Fig. S8a).

We added the data in revised Supplementary Fig. S8a and revised the text as follows;

“Therefore, we compared the expression patterns of INHBB and INHA in human cholesteatoma fibroblasts by ddPCR. The results showed that INHBA was significantly upregulated relative to INHBB and INHA in cholesteatoma fibroblasts (Supplementary Fig. S8a).” (page 14, line 5-8)

[Comment]

5. Fig 4c. In the transplantation experiments of Fig4c, the INHBA KO group resulted in fewer osteoclasts. Therefore, the authors suggest that INHBA/activin A from the pathogenic fibroblast was a key molecule for ectopic osteoclastogenesis. However, these experimental results can also be explained in other ways. For example, the IHNBA knockout can lead to the depletion of the fibroblast after transplantation. Alternatively, INHBA may form activin AB or inhibin A which make contribute to these results, since the author did not show the expression level of INHBB and INHA in mouse fibroblast. Additional experiments are necessary to be shown. Such as the pattern of transplanted fibroblast (which may form epidermal cysts according to the authors' previous work (Yoriko Iwamoto, 2015)). The expression level of INHBB and INHA in mouse fibroblast needs to be detected. The author also needs to check the RANKL expression level in the cultured fibroblasts that are ready to be injected.

[Response]

We appreciate the reviewer's important comment. As suggested by the reviewer, we compared the expression levels of *INHBA*, *INHBB*, and *INHA* in mouse cholesteatoma fibroblasts and mouse ear pinna fibroblasts. The gating strategy is shown in **revised Supplementary Fig. S10a**. *INHBA* expression was significantly upregulated in mouse cholesteatoma fibroblasts compared to mouse ear pinna fibroblasts (**revised Supplementary Fig. S10b**). On the other hand, *INHBB* and *INHA* expression levels were lower than the *INHBA* expression level in mouse cholesteatoma fibroblasts (**revised Supplementary Fig. S10c**). This suggests that *INHBA* forms dimers and activin A secretion is elevated in the mouse cholesteatoma model, and that activin A secretion may be decreased by *INHBA* knockout.

Next, we examined the RANKL expression in cultured fibroblasts that were ready to be injected and compared it to mouse ear pinna fibroblasts. RANKL expression was detected in the fibroblasts before injection, but not in mouse ear pinna fibroblasts (**Reviewer-only Fig. 3**), which was consistent with our previous report (Iwamoto *et al.*, *Mol Cell Biol*, 2016). These results suggest that RANKL expression in fibroblasts is required for osteoclastogenic function of activin A in the mouse cholesteatoma model.

Reviewer-only Figure 3

We added the data in **revised Supplementary Fig. S10** and revised the text as follows;

*“To elucidate the expression of *INHBA*, *INHBB*, and *INHA* in cholesteatoma mouse*

model fibroblasts, we collected fibroblasts from the cholesteatoma mouse model and control mouse ear pinnae via cell sorting (Supplementary Fig. S10a). The expression of INHBA was significantly upregulated in cholesteatoma mouse model fibroblasts, compared with controls (Supplementary Fig. S10b). Additionally, the levels of INHBB and INHA expression were lower than the level of INHBA in cholesteatoma mouse model fibroblasts (Supplementary Fig. S10c). These results suggested that INHBA forms dimers and activin A secretion is elevated in the cholesteatoma mouse model.” (page 16, line 4-12)

[Comment]

6. Fig4d. The positive control was provided as Control/FST- group. A negative control, such as the PBS injection group, is necessary for a better understanding of these results.

[Response]

As the reviewer requested, we have included PBS (vehicle)-injected group as a negative control for the FST-treated group in the revised manuscript (**revised Fig. 4f, g**).

We revised the data in **revised 4f, g** and rewrote the text as follows;

“Furthermore, application of the activin A antagonist follistatin (FST) suppressed osteoclastogenesis compared with the vehicle-injected group (Fig. 4f, g).” (page 17, line 2-3)

Response to Reviewer #3

[Comment]

This is a very impressive study that provides a cellular and molecular mechanism to explain why bone erosions occur in cholesteatoma. Using scRNA analysis of samples from four patients with cholesteatoma and four control samples the authors identify enrichment of a pathogenic subset of fibroblasts that express inhibinBA and with RANKL drive osteoclastogenic differentiation. Using both genetic and pharmacological inhibition of inhibinBA in fibroblasts they show that bone erosions are reduced in a mouse model of cholesteatoma.

While the authors have done a first-rate job in revealing a new and exciting mechanism, I have a few issues with the interpretation of their data a which I hope, if addressed, would make their findings even more interesting.

Major points

1. The nature of the control tissue (retro auricular skin) is important as all further comparisons of enrichment of populations in the cholesteatoma are made against this control. Skin is derived from ectoderm whereas the embryology of the middle ear requires many separate components from different embryonic origins and in particular form from the mesoderm of the branchial pouches. I am concerned that the comparison of cholesteatoma fibroblasts to skin fibroblast is not a fair one and a better one would be with mesenchymal fibroblast from for example synovial tissue. The auditory ossicles are connected by synovial joints so this would be a better control even if taken from a large synovial joint. What would happen to the analysis if the data in Fig 2c were a comparison against publicly available data sets of (for

example PMID: 35649411 or PMID: 31061532) synovial fibroblasts compared to their retro auricular skin skin (control) fibroblasts.

[Response]

We cordially thank the reviewer for the positive and instructive comments. We agree that it is better to use mesenchymal fibroblasts, such as synovial fibroblasts, as controls for the single-cell RNA-seq analysis. Synovial membranes of the ossicular bones may be used, but are limited in quantity. In addition, cholesteatomas expand in the middle ear, so the ossicular bones could not be used for analysis. Healthy ossicular bones are necessary for hearing, therefore, the healthy ossicular bones could also not be used. We could not obtain synovial membrane from other healthy organs because it is ethically impermissible to harvest normal tissue from non-disease-related sites. Therefore, the only normal tissue accessible during cholesteatoma surgery was the retroauricular skin at the incision site, which could be removed during the surgery without causing additional morbidity. Although we used different tissues for single-cell RNA-seq, the scRNA-seq results showed that the two tissues had similar cell types.

We also performed integrated data analysis for the single-cell RNAseq data with the public datasets. Since the suggested public datasets (PMIDs: 35649411 and 31061532) included synovial membranes from the joints of rheumatoid arthritis or osteoarthritis patients, we integrated our dataset with the cross-tissue, single-cell, stromal atlas data (PMID: 35649411). The expression profiles that were common between our dataset and the public dataset were normalized, clustered, and integrated through the ingest function of Scanpy (**Reviewer-only Fig. 4a**). The integrated data analysis revealed that the fibroblasts in our study had molecular properties similar to the fibroblasts collected from synovial membranes of rheumatoid arthritis joints (**Reviewer-only Fig. 4b, c**). The

similar molecular properties of cholesteatoma fibroblasts and synovial membrane fibroblasts from rheumatoid arthritis joints reflect their common characteristics as inflammatory bone destructive diseases.

It has been previously reported that fibroblast-like synoviocytes express activin A, and that disease severity is correlated with circulating activin A levels (El-Gendi *et al.*, *Int J Rheum Dis*, 2010). It has also been reported that fibroblasts are a source of activin A in the mouse arthritis model (Waltereit-Kracke *et al.*, *Ann Rheum Dis*, 2022). The present study was the first to identify an activin A-producing osteoclastogenic fibroblast subset in cholesteatoma patients. We hypothesize that activin A may be a key molecule for inflammatory bone destruction.

Reviewer-only Figure 4

Based on the discussion, we added the following description;

“To explore fibroblast similarity between cholesteatoma and rheumatoid arthritis, we compared data from the present study with cross-tissue, single-cell stromal atlas data³⁹. We downloaded UMI matrix and metadata from the Single Cell Portal (Broad Institute). The expression profiles of genes shared with the scRNA-seq data from the present study

were normalized, clustered, and integrated using the *ingest* function of Scanpy. Mapping onto the cross-tissue stroma atlas data revealed that the fibroblasts described in our study had molecular properties similar to the properties of fibroblasts collected from the synovial membrane of joints in rheumatoid arthritis. These results indicated that the molecular properties of fibroblasts in our study were similar to the properties of fibroblasts in the synovial membrane of joints in rheumatoid arthritis, suggesting that fibroblasts have common characteristics in cholesteatoma and rheumatoid arthritis—both constitute inflammatory bone destructive diseases.” (page 21, line 1-10)

[Comment]

2. I do not understand why the authors use cluster 2 as the common origin for cholesteatomata and control (skin) fibroblasts. Does this cluster contain P16 the recently described marker for universal fibroblasts (PMID: 33981032). If it does then I could understand the choice of cluster 2 for the pseudotime trajectory. If not, it seems a rather random choice to start the trajectory analysis from cluster 2. What other genes in subset 2 make the authors think this is a common origin fibroblast?

[Response]

We thank the reviewer’s significant comment. As the reviewer suggested, we confirmed the P16 expression in our single-cell RNA-seq dataset. Since *PI16* was most highly expressed in subcluster 9 (**revised Supplementary Fig. 5b**), we decided to use it as the trajectory root and performed the pseudotime trajectory analysis. We also found that subcluster 8, the choleseatoma-specific fibroblast, differentiated from subcluster 9 via subclusters 3, 2, 10, 7, and 11 (**revised Fig. 3b**).

We added the data in **revised Supplementary Fig. S5b** and revised the text as follows;

“To infer the process of development from normal to disease-specific fibroblasts, we classified each fibroblast cluster into smaller subclusters and performed pseudotime trajectory analysis using Monocle 3 (Fig. 3a, b). Subcluster analysis revealed that the

*fibroblasts contained 15 subclusters. Cholesteatoma-specific fibroblasts were divided into five subclusters designated as 1, 7, 8, 10, and 11 (Fig. 3a). We identified the top 10 marker genes indicating the differentiation status of fibroblasts, then selected the P116 gene (described as a universal fibroblast marker in a previous study¹⁵) (Supplementary Fig. S5a). Subcluster 9 was defined as the origin of fibroblasts for pseudotime trajectory analysis because cells in this subcluster showed high levels of P116 expression (Supplementary Fig. S5b). Pseudotime trajectory analysis suggested that the cholesteatoma-specific fibroblasts differentiated from subcluster 9 toward subcluster 8, via subclusters 3, 2, 10, 7, and 11 (Fig. 3b). The marker gene P116 was downregulated after differentiation (Supplementary Fig. S5b). Subcluster 8 was composed of the most differentiated cholesteatoma-specific fibroblasts and exhibited markedly higher *INHBA* expression level than the other subclusters in the cholesteatoma-specific fibroblasts (Fig. 3b–d, Supplementary Fig. S5c).” (page 10, line 4–page 11, line 3)*

Minor points

[Comment]

3. Why are mouse ear pinna derived fibroblasts used in Fig 3g and not human retro auricular skin fibroblasts.

[Response]

As the reviewer suggested, we performed the differentiation assay with human skin primary fibroblasts. IL1- β , PGE₂, and TNF- α promoted *INHBA* expression in human skin fibroblasts (**revised Fig. 3g**). This was consistent with the previous results in mouse fibroblasts.

[Comment]

4. In the discussion (line 245-248) the authors mention that infiltrating CD45+ cells produce proinflammatory cytokines and PGE2. Do they think these are macrophages and/ or granulocytes? If so could these cells be added to Fig 5 as the

source of these factors.

[Response]

As stated by the reviewer, it has previously been reported that the proportion of M1 macrophages producing inflammatory cytokines is elevated in cholesteatomas (Bassiouni *et al.*, *J Clin Med*, 2022). Therefore, we have revised **Fig. 5** accordingly.

We revised the text as follows;

“Additionally, we reported previously that infiltrating mononuclear cells express IL-1 β in human cholesteatoma⁴ and previous research reported that the proportion of M1 macrophages producing inflammatory cytokines is elevated in cholesteatoma²⁵. Taken together, our observations suggest that infiltrating CD45⁺ cells, particularly macrophages, produce proinflammatory cytokines, thereby promoting pathogenic fibroblast differentiation and activin A secretion that result in increased ectopic osteoclastogenesis in human cholesteatoma.”

(page 19, line 6-12)

REVIEWERS' COMMENTS

Reviewer #1 (Remarks to the Author):

The revised manuscript " Single-cell transcriptomics of human cholesteatoma identifies an activin A-producing osteoclastogenic fibroblast subset inducing bone destruction" is acceptable in the current form. All the points have been addressed!

Reviewer #2 (Remarks to the Author):

The authors have addressed all of the points arising from my initial review of the original manuscript.

Reviewer #3 (Remarks to the Author):

Thank you for the careful response to my two main points. I still have major reservations that these critical points have not been addressed

1. The nature of the control tissue

The new comparison of cholesteatoma fibroblasts to synovial fibroblasts compared to skin has significantly changed the interpretation of the manuscript. The integrated data analysis has revealed that the cholesteatoma fibroblasts are remarkably similar to synovial fibroblasts (Reviewer-only Fig. 4b, c). This reduces the novelty of the findings

2. Origin of the cholesteatoma fibroblasts

I do not think that this issue has been adequately addressed and in the absence of adequate tissue localization (spatial transcriptomics) may turn out to be speculation rather than ground truth

Response to the Reviewers

We thank the reviewers for the positive and constructive comments about our manuscript, "**Single-cell transcriptomics of human cholesteatoma identifies an activin A-producing osteoclastogenic fibroblast subset inducing bone destruction**" (NCOMMS-22-47016A), by Shimizu *et al.* We revised the manuscript in response to the comments. We hope that we have responded adequately and that the revised manuscript is now suitable for publication.

Response to Reviewer #1

[Comment]

The revised manuscript " Single-cell transcriptomics of human cholesteatoma identifies an activin A-producing osteoclastogenic fibroblast subset inducing bone destruction" is acceptable in the current form. All the points have been addressed!

[Response]

We thank the reviewer for the positive comment regarding our revised manuscript.

Response to Reviewer #2

[Comment]

The authors have addressed all of the points arising from my initial review of the original manuscript.

[Response]

We thank the reviewer for the positive comment regarding our revised manuscript.

Response to Reviewer #3

[Comment]

Thank you for the careful response to my two main points. I still have major reservations that these critical points have not been addressed

1. The nature of the control tissue

The new comparison of cholesteatoma fibroblasts to synovial fibroblasts compared to skin has significantly changed the interpretation of the manuscript. The integrated data analysis has revealed that the cholesteatoma fibroblasts are remarkably similar to synovial fibroblasts Reviewer-only Fig. 4b, c). This reduces the novelty of the findings

[Response]

We appreciate the reviewer's additional comments. The reviewer rightly highlights that the integrated data analysis with various public datasets of fibroblasts reveals a gene expression profile in cholesteatoma fibroblasts similar to that of fibroblasts in rheumatoid arthritis. However, we do not think that this undermines the novelty and significance of the present study. The major points demonstrated in this study are; 1) the identification of a subset of pathogenic fibroblasts in cholesteatoma that expresses activin A, and 2) the concept that the induction of osteoclasts mediated by these pathogenic fibroblasts would be a shared mechanism not only in cholesteatoma but also in such diseases involving bone destruction as rheumatoid arthritis. The integrated analysis strongly supports this concept of shared mechanism underlying the induction of bone destruction. This could lead to the identification of a shared therapeutic target across such diseases.

We are immensely grateful to the reviewer for suggesting the integrated analysis with

public databases, which indeed strengthens our concept. Consequently, we have included the previously reviewer-only Figure 4 into Supplementary Figure S11d-f and have discussed it in the Discussion section.

“Mapping onto the cross-tissue stroma atlas data indicated that the molecular properties of fibroblasts in our study were similar to the properties of fibroblasts in the synovial membrane of joints in rheumatoid arthritis, suggesting that fibroblasts have common characteristics in cholesteatoma and rheumatoid arthritis—both constitute inflammatory bone destructive diseases (Supplementary Fig. S11 d–f). Taken together, our results support the suggestion that activin A is a common key molecule for ectopic osteoclastogenesis and subsequent pathological bone destruction in inflammatory diseases.” (page 20, line 307-314)

2. Origin of the cholesteatoma fibroblasts

I do not think that this issue has been adequately addressed and in the absence of adequate tissue localization (spatial transcriptomics) may turn out to be speculation rather than ground truth

[Response]

We are sincerely grateful to the reviewer for the insightful comments. We concur with the viewpoint regarding the need for further analysis to discern the origin of the cholesteatoma fibroblasts. We found that *PII6* expression was most pronounced in subcluster 9. However, due to challenges in accessing human specimens and obtaining suitable antibodies for staining, we were unable to include additional analyses such as tissue section staining or spatial transcriptomics in the revised manuscript. We have now addressed these limitations pertaining to the origin of the fibroblasts in the Results and

Discussion sections.

“The marker gene P116 was downregulated after differentiation (Supplementary Fig. S5b). These results suggest subcluster 9 as the potential origin of fibroblasts; however, a comprehensive tissue localization analysis and lineage tracing are necessary to definitively establish the source of fibroblasts.” (page 10, line 148-line 151)

“While subcluster 9 is hypothesized as the origin of fibroblasts, additional lineage tracing and comprehensive tissue localization analysis will be required in future studies to definitively establish the source of fibroblasts.” (page 19, line 289-291)